# Marine Flora of French Polynesia: An Updated List Using DNA Barcoding and Traditional Approaches

**DOI:** 10.3390/biology12081124

**Published:** 2023-08-11

**Authors:** Christophe Vieira, Myung Sook Kim, Antoine De Ramon N’Yeurt, Claude Payri, Sofie D’Hondt, Olivier De Clerck, Mayalen Zubia

**Affiliations:** 1Department of Biology and Research Institute for Basic Sciences, Jeju National University, Jeju 63243, Republic of Korea; myungskim@jejunu.ac.kr; 2Phycology Research Group, Center for Molecular Phylogenetics and Evolution, Ghent University, 9000 Ghent, Belgium; 3Pacific Center for Environment an Sustainable Development, The University of the South Pacific, Private Mail Bag, Suva P.O. Box 1168, Fiji; nyeurt@gmail.com; 4Institut de Recherche pour le Développement, Nouméa 98848, New Caledonia; 5UMR Ecosystèmes Insulaires Océaniens, University of French Polynesia, BP6570, Faa’a 98702, Tahiti, French Polynesia

**Keywords:** algae, Chlorophyta, Cyanobacteria, Rhodophyta, molecular-assisted alpha taxonomy, Ochrophyta, seagrass, seaweeds

## Abstract

**Simple Summary:**

The French Polynesian islands represent a unique insular system in the Pacific Ocean. Previous surveys of the marine flora of French Polynesia were mostly established on traditional morphology-based taxonomy. DNA barcoding allowed us to provide a major revision of French Polynesian marine flora, with an updated total of 702 species from French Polynesia, including 119 species of Chlorophyta, 169 Cyanobacteria, 92 Ochrophyta, 320 Rhodophyta, and 2 species of seagrasses (Alismatales)—nearly a two-fold increase from previous estimates. In addition to improving and refining our knowledge of French Polynesian marine flora, this study also provides a valuable DNA barcode reference library for identification purposes and future taxonomic and conservation studies. A significant part of the diversity uncovered corresponds to unidentified lineages, which will need careful taxonomic examination.

**Abstract:**

Located in the heart of the South Pacific Ocean, the French Polynesian islands represent a remarkable setting for biological colonization and diversification, because of their isolation. Our knowledge of this region’s biodiversity is nevertheless still incomplete for many groups of organisms. In the late 1990s and 2000s, a series of publications provided the first checklists of French Polynesian marine algae, including the Chlorophyta, Rhodophyta, Ochrophyta, and Cyanobacteria, established mostly on traditional morphology-based taxonomy. We initiated a project to systematically DNA barcode the marine flora of French Polynesia. Based on a large collection of ~2452 specimens, made between 2014 and 2023, across the five French Polynesian archipelagos, we re-assessed the marine floral species diversity (Alismatales, Cyanobacteria, Rhodophyta, Ochrophyta, Chlorophyta) using DNA barcoding in concert with morphology-based classification. We provide here a major revision of French Polynesian marine flora, with an updated listing of 702 species including 119 Chlorophyta, 169 Cyanobacteria, 92 Ochrophyta, 320 Rhodophyta, and 2 seagrass species—nearly a two-fold increase from previous estimates. This study significantly improves our knowledge of French Polynesian marine diversity and provides a valuable DNA barcode reference library for identification purposes and future taxonomic and conservation studies. A significant part of the diversity uncovered from French Polynesia corresponds to unidentified lineages, which will require careful future taxonomic investigation.

## 1. Introduction

### 1.1. French Polynesia

French Polynesia consists of strings of islands positioned in the heart of the South Pacific Ocean, arranged into five archipelagos: the Society, Marquesas, Austral, Gambier, and Tuamotu Islands [1] (Figure 1). French Polynesian island chains resulted from linear volcanic hotspot activities on the Pacific plate that began during the Eocene (47.4 Ma, Tuamotu archipelago [2]; but see [3]). The five archipelagos feature some 120 emerged islands and more than 500 seamounts [4], dispersed over an area of 2,500,000 km^2^ [5], forming a total land area of 3521 km^2^ with a combined coastline of 5830 km [6]. French Polynesian emerged islands present a variety of stages of the tropical volcanic island life cycle, ranging from high islands (such as Tahiti and Moorea) to atolls (such as Rangiroa and Tikehau) and guyot (or seamount), with variable climates linked to their geographical position. French Polynesia is the world’s largest contiguous exclusive economic zone (EEZ) with a total of 5,030,000 km^2^ [7].

### 1.2. Relatively Poor Biodiversity

Chiefly imputable on the insularity, remoteness, and smallness of its islands, French Polynesia supports a relatively low terrestrial and marine biodiversity [8,9,10,11]. Some authors noted that due to geographic isolation, some algal families were poorly represented in French Polynesia [12]. Contrasting to the absence of particular families, levels of endemism in some seaweed groups can be remarkably high [9,13]. In comparison to their terrestrial counterpart (893 indigenous plant species and 1700 introduced species [9]), the marine flora biodiversity is significantly poorer. The latest published works on French Polynesian marine flora reported a total of 332 species of marine macroalgae (198 Rhodophyta, 32 Ochrophyta, 82 Chlorophyta [14,15,16]), 117 Cyanobacteria [17], and 2 seagrasses [18]. Since then, only a limited number of taxonomic studies have been conducted in French Polynesia, e.g., [13,19,20,21].

### 1.3. Phycological History of French Polynesia

The earliest recorded collections of marine algae from French Polynesia were made during exploration voyages in the 18th and 19th centuries, and these were later examined and published by several phycologists in the 19th and early 20th centuries [22,23,24,25,26,27,28,29,30,31,32,33,34,35,36,37,38,39,40]. The first major floristic account of Tahitian algae was by Setchell [41], listing and illustrating 201 taxa of marine and freshwater algae. Hollenberg [42,43,44,45] published on species of Rhodomelaceae collected by M. S. Doty and J. Newhouse in the early 1950s from the Tuamotu archipelago; in 1973, W. R. Taylor reported on algae collected during the Smithsonian–Bredin Expedition to the Society and Tuamotu Islands [46]. Working on Tahitian collections by Setchell and Parks in 1922 and Crossland in 1928–1929, housed at the University of California Herbarium (UC), Olsen-Stojkovich [47] published a dissertation on the genus *Avrainvillea*. The same year, Payri and Meinesz [48,49] listed 346 taxa of French Polynesian algae, followed by Abbott [50], who reported on a new species of *Valoniopsis* from Huahine Island. The majority of earlier works on French Polynesian marine plants were regrouped in a comprehensive checklist by Payri and N’Yeurt [17], which also included species from Moorea Island reported by Payri [51] in her doctoral thesis dissertation. A richly illustrated field guide to French Polynesian marine flora directed at the general public was published by Payri, N’Yeurt, and Orempüller [52]. At this time, the University of French Polynesia’s phycological herbarium (officially recognized as UPF) was created by Antoine De Ramon N’Yeurt and Claude Payri to house the growing number of algal specimens stemming from ongoing work on French Polynesian flora. These efforts culminated in the publication of a three-part scientific marine flora for French Polynesia, detailing the Ochrophyta, Chlorophyta, and Rhodophyta [14,15,16]. Three new species of Rhodophyta from the Marquesas and Tuamotu archipelagos resulting from this work were published in [53]. The number of endemic species identified from the region, prior to molecular studies, appears remarkably low.

### 1.4. Molecular-Based and Barcoding Taxonomic Studies

A limited number of molecular-assisted alpha-taxonomic studies have been conducted in French Polynesia, encompassing the Rhodophyta (Gelidiales [54], *Gibsmithia* [55], *Asparagopsis* [56], *Grateloupia filicina* [57,58], Corallinophycidae [59,60,61,62], Laurencieae [63]), Chlorophyta (*Halimeda* [64,65,66,67,68]), Ochrophyta (*Dictyota* [69,70,71,72], *Lobophora* [13,73], *Newhousia* [21], *Sargassum* [74], *Turbinaria* [75]), and Cyanobacteria [20], but nothing on seagrasses. These molecular-based studies generated over 500 sequences, but from a limited number of taxa, and a bulk from the genus *Lobophora* (>350 sequences [13]), followed by *Halimeda* (>90 sequences [64,65,66,67,76]), *Sargassum* (>30 sequences [74]), and *Newhousia* (20 sequences [21]). These studies included several markers, encompassing *cox*1, *cox*2-*cox*3 intergeneric spacer, *cox*3, *nad*1, *psb*A, *psa*A, *rbc*L, *tuf*A, ITS, 16S, 18S, 28S, 23S, 26S. Between 2008 and 2010, the Moorea BIOCODE Project led by the University of California at Berkeley in collaboration with the Smithsonian National Museum of Natural History listed a comprehensive inventory of all non-microbial life in a reef-to-ridge transect of Moorea Island, Society Islands, French Polynesia [77]. While sequences were successfully generated for organisms such as corals and fish (e.g., [78]), we could not find any useful sequences (i.e., a BLAST search returned no match to any sequences in the genomic BLAST database on NCBI BLAST) for the 28 marine algae (27 Rhodophyta and one Chlorophyta) in the publicly available dataset [79]. We suspect that the markers used for the sequences were not appropriate for marine algae.

### 1.5. DNA Barcoding Approach

There are several challenges to a DNA barcoding approach that have been discussed in [80], including the completeness and reliability of the reference library (e.g., non- and mis-labeled sequences) and the limited resolution of some markers. Gene libraries are often biased towards temperate regions, potentially neglecting tropical biodiversity in poorly sampled areas. There is an extensive mislabeling on GenBank that results in numerous polytomic species names. It is, therefore, a priority to augment the available GenBank records with high-quality sequences from all biogeographic areas, providing as much metadata as possible along with the sequences, while avoiding assigning names to submissions that do not align with any known sequences.

### 1.6. Barcoding an Entire Regional Marine Flora

Previous efforts of DNA barcoding of marine flora have focused either on (1) restricted geographic areas covering the three main macroalgal groups (Chlorophyta, Phaeophyceae, and Rhodophtya; e.g., northern Madagascar [80], Bergen, Norway [81], Boulder Patch, Beaufort Sea [82], southeast coast of India [83], Malta [84]), (2) larger areas (e.g., regional scale) but targeting specific groups (e.g., Rhodophyta in South Africa [85,86] and Qingdao, China [87]; Dumontiaceae [88] and Phyllophoraceae [89] in Canada; Rhodymeniales in Australia [90]; Gracilariaceae [91] and *Pyropia* [92] in the Republic of Korea), or (3) on lower taxonomic levels (e.g., genus, order, or family) but covering large geographic scales (e.g., Corallinophycidae in Altantic European maerl beds [93]). To our knowledge, no DNA barcoding study has yet been conducted at a regional level encompassing the three macroalgal groups (Chlorophyta, Ochrophyta, Rhodophyta), Cyanobacteria, and the marine phanerogams (seagrasses; marine phanerogams).

### 1.7. Objective of the Study

The present study aimed to re-assess the species diversity of French Polynesian marine flora using DNA barcoding and to deliver a revised checklist for the marine flora of French Polynesia including three macroalgal classes (Rhodophyta, Ochrophyta, Chlorophyta), the Cyanobacteria (Cyanobacteria), and the seagrasses (Alismatales).

## 2. Materials and Methods

### 2.1. Study Site and Sampling

This study was conducted between 2014 and 2023. Marine floral specimens were collected in the subtidal zone via SCUBA or snorkeling and in the shallow intertidal zone on foot and through wading or snorkeling. Specimens identified by Antoine De Ramon N’Yeurt using morphological methods while working with the Moorea BIOCODE Project in 2008 were also included in the present study.

Specimens were initially identified at the genus level and, when possible, at the species level based on the morphology. In situ and ex situ photographs were taken of most specimens collected. After field collection, for each taxa, specimens were (1) mounted as herbarium vouchers, (2) fixed in formaldehyde (4% formaldehyde in seawater) for anatomical examination, (3) and small fragments (~1 cm^2^) were preserved either in fine silica gel (~0.2–1 mm; ref. 1.01905.1000; Merck KGaA, Damstadt, Germany) or in ethanol (absolute EtOH) for molecular analyses. For the macroalgae, morphospecies determination was done using monographs and identification manuals from the region [14,15,16] as well as based on the most recent publications for the identified taxa. Detailed morphological and anatomical examinations and final identifications were carried out at the University of French Polynesia using a binocular and a light microscope D2000 (Leica Microsystems, Wetzlar, Germany), equipped with a Canon EOS 600D. For the Cyanobacteria, morphological observations consisted of microscopic, ultrastructural, and morphometric analyses. Fresh samples of Cyanobacteria were examined immediately following collection using a light microscope DMZ50 (Leica Microsystems, Wetzlar, Germany) to locate representative subsamples and taxonomically uniform colonies. Fresh and semi-permanent slides were prepared. Observations and measurements were made using a light microscope D2000 (Leica Microsystems, Wetzlar, Germany), equipped with a Canon EOS 600D. Measurements were carried out with Sigma-Scan Image analytical software (Sausalito, CA, USA) and Motic Images Plus (Motic Group, Hong Kong, China), using a calibrated ocular micrometer and in-scale projections and photomicrographs. Phenotype determination was performed using the available monographs and identification manuals (see [94]) as well as more recent publications, e.g., [95,96,97,98,99,100]. Some taxonomic revisions with designation changes were introduced following our phylogenetic reconstructions. A voucher number was assigned to each sample together with the date of collection and deposited into the Phycological Herbarium of the University of French Polynesia (UPF). A total of 2435 (1823 macroalgae + 612 Cyanobacteria + 10 seagrasses) specimens were collected (Appendix A).

### 2.2. DNA Extraction, PCR Amplification and Sequencing

Total genomic DNA was extracted from tissue samples dried in silica gel using either (1) a cetyl-trimethyl ammonium bromide extraction method [101] or (2) the MagPurix^®^ Plant DNA Extraction Kit v.1.3 (ZP02014; Zinexts Life Science Corporation, New Taipei City, Taiwan) according to the manufacturer’s instructions.

For the CTAB protocol, the silica-dried portion was ground in liquid nitrogen using a mortar and pestle. Extraction of total genomic DNA was carried out using the protocol from OmniPrep for plant tissue (G-Biosciences, St. Louis, MO, USA). For the MagPurix protocol, algal material was directly processed, without grinding, within PLA buffer, and incubated for 4 h at 60 °C. The MagPurix DNA Extraction Kit was run on the MagPurix 12A automated nucleic acid extraction system (Zinexts Life Science Corporation, New Taipei City, Taiwan). The DNA extract (final volume of 100 µL) was stored at −24 °C.

Sequences were generated from the chloroplast ribulose-bisphosphate carboxylase gene (*rbc*L) (for the Rhodophyta, Ochrophyta, Chlorophyta), the chloroplast-encoded elongation factor Tu (*tuf*A) (Chlorophyta), the plastid-encoded PSII reaction center protein D1 (*psb*A) (Rhodophyta, Ochrophyta), the mitochondrial-encoded cytochrome oxidase subunit 3 (*cox*3), (*cox*1) (Rhodophyta, Ochrophyta, Chlorophyta), 18S (Chlorophyta), 28S (Chlorophyta), 16S rRNA (Cyanobacteria), ITS1 (Alismatales) (Appendix A).

The primer pairs for the amplification and sequencing of each gene were as follows: for *cox*1 M13LF3-M13Rx (Rhodophyta) [102], GazF1-GazR1 [102], for *cox*3, CAF4A-CAR4A (Ochrophyta) [103], and *cox*3_44F-*cox*3-739R (Ochrophyta) [104], for *rbc*L 68F-R708 [105,106], PRB-F0-PRB-R1A, PRB-F2-PRB-R2, PRB-F3-PRBR3A, *rbc*L3F-RSPS (Ochrophyta) [107], DRL1F-DRL1R (Ochrophyta) and DRL2F-DRL2R (Ochrophyta) [108], 3F-S97R (Ochrophyta) [109], and F57-*rbc*LrevNEW (Rhodophyta) [102,110], F7-RrbcS start (Rhodophyta) [110], G*rbcL*nF-G*rbc*LR (Chlorophyta) [111], for *psb*A *psb*A_F-*psb*A_R1 [112], for *tuf*A *tuf*AF-*tuf*AR or tufG4F-*tuf*AR [111,113], for 28S rRNA C’1-D2 or C’2B-D2 (Chlorophyta) [114], for 16S rRNA (Cyanobacteria) [115], and for ITS P674-P675 [116,117] (Alismatales) (Appendix A).

PCRs were either carried out in tube strips in 20 µL reaction volumes using the following PCR master mix: (1) the MG 2X Taq PreMix (MGmed, Seoul, Republic of Korea), (2) the AccuPower^®^ Taq PCR Premix kit (Bioneer Corp., Daejeon, Republic of Korea); or in 96-well plates in 25 μL reaction volumes containing 10X PCR buffer (2.5 µL), 200 μM dNTPs (2.5 µL), 10 µM of each primer (1.25 µL), 10µg/µL BSA (1 μL), 10 ng (1 μL) of genomic DNA, and 1 U/µL of TaqDNA polymerase (Amplitaq DNA polymerase, N8080152, Thermo Fisher, Seoul, Republic of Korea). PCRs were carried out using a thermocycling profile with specific parameters for each marker and primer set used, indicated in Appendix A. PCRs were run either in a (1) Kyratec (SC300G-R2; Kyratec, Mansfield, Australia), (2) AllInOneCycler 96-well PCR system (A-2041-1 N; Bioneer Corp., Daejeon, Republic of Korea), (3) Biometra T-professional 96 (BM070-701, Westburg, The Netherlands), or (4) VerityTM (4375786, Thermo Fisher). Sequencing reactions and runs were performed by Macrogen (Seoul, Republic of Korea, or Amsterdam, The Netherlands).

### 2.3. Sequence Alignment, Phylogenetic Reconstruction, Molecular-Assisted Taxonomic Identification

Nucleotide sequences newly generated were firstly BLASTed against the genomic BLAST database on NCBI BLAST (Basic Local Alignment Search Tool). BLASTing results allowed for the confirmation that newly generated sequences originated from algal DNA material (and not from microbial contaminants or epiphytes) and were more or less in line with preliminary field identification. All nucleotide sequences available on NCBI GenBank were downloaded for the corresponding marker and taxon, either at the generic level or at a higher taxonomic level (e.g., family level), if doubt on the generic identification existed. Several names assigned to sequences deposited on GenBank are inaccurate or were deposited in GB with a preliminary/different species label prior to final publication) and have not been updated since their publications. Sequences names from GenBank were therefore curated whenever possible based on the most up-to-date taxonomic information available on AlgaeBase (e.g., through updating sequences names with currently accepted species names) and the latest molecular taxonomic studies (e.g., those that delivered amendments on species names assigned to lineages). For large datasets (e.g., greater than 200 sequences) we kept a single representative sequence per haplotype, defined as >99.9% similar, using the CD-HIT program [118] run on a local computer.

Nucleotide sequences newly generated in this study were added to the GenBank downloaded sequence datasets and aligned using MUSCLE v.3.5 [119], with default parameters implemented in the eBioX software package v.1.5.1 [120].

Maximum-likelihood phylogenetic trees were reconstructed from each marker and each class using a best fit substitution model and an SPR branch swapping algorithm in PhyML v.3.0 [121].

Molecular-assisted taxonomic identification was based on placement of newly generated sequences within the phylogenetic trees and sequence similarity. Whenever newly generated sequences did not position clearly within a clade (i.e., a cluster of sequences corresponding to a given species) and diverged from the closest sequences by >1%, these lineages were considered as unidentified species and given the specific epithet “sp.#FP”. Finally, we returned to the morphological observations to ensure that molecular-assisted taxonomic identification corresponded to morphological data, at least at the genus level.

## 3. Results

### 3.1. Sequence Data

A total of 2452 specimens (1823 macroalgae + 619 Cyanobacteria + 10 seagrasses) of the French Polynesian marine algal collection were processed, including sequences already published but part of the same projects on French Polynesian marine floral biodiversity (ALGALREEF, CYANODIV, CARISTO, MICROALG). Genomic DNA was extracted from a total of 2452 specimens, followed by PCR analyses. Finally, a total of 1007 sequences (783 macroalgae + 223 Cyanobacteria + 1 seagrass) were generated from 2452 specimens: 252 sequences for the Ochrophyta, including 29 *cox*1 sequences (Ochrophyta), 87 *cox*3 sequences (Ochrophyta), 76 *psb*A sequences (Ochrophyta), and 60 *cox*1 sequences (Ochrophyta); 262 sequences for the Chlorophyta, including 29 18S rDNA sequences (Chlorophyta), 41 28S rDNA sequences (Chlorophyta), 4 ITS sequences (Chlorophyta), 4 *rbc*L sequences (Chlorophyta), and 184 *tuf*A sequences (Chlorophyta); 270 sequences for the Rhodophyta, including 270 *rbc*L sequences (Rhodophyta); 1 sequence for the Alismatales, including 1 ITS sequence (Alismatales); 223 sequences from Cyanobacteria, including 61 [20] + 164 (new) 16S rRNA sequences (Cyanobacteria). Additionally, we recovered sequences from GenBank that were previously published, either focusing directly on French Polynesia (e.g., *Lobophora, Newhousia, Sargassum, Turbinaria* [13,21,74,75]) or not (e.g., *Dictyota* [122], *Halimeda* [65]). We retrieved a total of 585 sequences, including 69 sequences of Rhodophyta (25 *cox*1 sequences, 9 *rbc*L, 14 28S, 7 *psb*A, 4 LSU, 3 18S, 5 SSU, 2 23S), 466 sequences of Ochrophyta (1 26S, 7 *cox*1, 217 *cox*3, 1 *psa*A, 111 *psb*A, 119 *rbc*L, 1ITS, 3 *nad*1, 4 SSU, 1 trnW-trnI, 1 *tuf*A), and 50 sequences of Chlorophyta (*tuf*A only).

### 3.2. Species Identification

Based on molecular analyses, we have identified a total of 352 lineages/species: 1 Alismatales, 53 Cyanobacteria, 150 Rhodophyta, 77 Ochrophyta, and 71 Chlorophyta. We could confirm a total of 80 names from previous checklists (7 Cyanobacteria, 26 Rhodophyta, 14 Ochrophyta, 33 Chlorophyta). Our molecular phylogenetic study disclosed a total of 227 new lineages/species for French Polynesia: 115 Rhodophyta, 30 Chlorophyta, 61 Ochrophyta, 20 Cyanobacteria, 1 Alismatales. Among these 227 new lineages, 75 matched a name through the barcoding approach, and 152 did not (i.e., unidentified species) (113 Rhodophyta, 16 Chlorophyta, 23 Ochrophyta, 37 Cyanobacteria). Lineages labelled as “sp.#FP” diverged from the closest sequences from GenBank by >1%. It should be noted that these lineages do not necessarily correspond to new species, but from a barcoding approach, it strictly implies these sequences do not match sequenced species for these particular markers. Further analyses are needed to determine if these lineages should be defined as new species. Also, for those that match a current name, one needs to verify if the identifications of the names in GenBank are accurately based on type specimens or on secondary collections that could have been misidentified. Hence, these numbers need to be taken with caution at this time. We did not get molecular confirmation for 356 names from the previous checklist (170 Rhodophyta, 48 Chlorophyta, 15 Ochrophyta, 115 Cyanobacteria).

## 4. Discussion

This study aimed to revisit the marine floral biodiversity of French Polynesia through DNA barcoding. We discuss the present marine floral biodiversity findings and the value, utility, and challenges of barcoding an entire regional marine flora. Some of the species collected have been illustrated in Figure 2.

### 4.1. French Polynesian Marine Floral Species Diversity

Until the last update on regional flora from French Polynesia by N’Yeurt and Payri [16], the marine floral diversity consisted of 430 species: 198 Rhodophyceae, 32 Ochrophyta, 83 Chlorophyceae, and 117 Cyanobacteria. As mentioned earlier, a limited number of molecular-based studies have been conducted in French Polynesia in the past. Taking into account the diversity documented in past works on the French Polynesian flora and the diversity uncovered in the current study, we provide the numbers of 702 species, 320 Rhodophyta, 119 Chlorophyta, 92 Ochrophyta, and 169 Cyanobacteria. Moreover, it is very likely that an important fraction of the names previously documented and not verified with the barcoding approach are misapplied names due to the limitations of traditional methods in resolving morphologically identical species. Previous molecular-based studies have already confirmed this with some pan-tropical genera: *Lobophora* [13], *Gibsmithia* [55], and *Sargassum* [74]. The case of the genus *Lobophora* perhaps illustrates this best; previous works provided the widely-distributed name *Lobophora variegata*, which in fact corresponds to 37 different pseudo-cryptic species that would have been very difficult to discern using traditional techniques [123]. At the family level, e.g., Scytosiphonaceae, among the six previously identified names, four were confirmed via barcoding, six were new, and another two have not been confirmed via barcoding, leaving their previous identification as disputable.

From our study (Table 1), the level of marine floral species endemism is 11% (only including currently accepted names) and extends to 28% when accounting for all new lineages (i.e., unidentified species). These results, however, need to be taken with caution, as floristic species richness and single-island endemic species richness can be unknown, with dynamic figures dependent on sampling effort [124]. This is especially true of French Polynesia, with its vast geographical area and diversity of habitats.

### 4.2. Matching Barcode Data to Morphological-Based Identifications

Matching sequences to previously identified names turned out to be more challenging than expected. We were able to match sequences onto only 80 names (~18.6%) of the 430 names that were previously identified based on morphological data only. In the process, we unveiled a large potentially unknown species diversity (227 new lineages/species), including 75 previously described species and 152 unknown lineages. Accordingly, important taxonomic work remains to be conducted on French Polynesian marine flora to confirm if these new lineages and species reported through barcoding represent actual new taxa. The Rhodophyta in particular will require extensive studies, with no less than 113 unmatched lineages, i.e., not closely matching sequences from GenBank for our reference markers. This work would be all the more important considering that biogeographically speaking, species diversity generally decreases as one moves east from the Indo-Pacific centers of distribution [13,128,129].

### 4.3. Challenges of Barcoding an Entire Marine Flora

Barcoding an entire marine flora can be a complex and challenging task, with several obstacles to take into account. Difficulties include the availability of a comprehensive barcode reference library, a lack of new specimens for rare or seasonal species, a lack of taxonomic expertise, and difficulties in the amplification and matching of barcode data [130]. Based on our own study, we have highlighted some of the major challenges we have faced when barcoding an entire flora that one should be aware of when conducting such studies. These difficulties are inherent to (1) data collection, (2) molecular analyses, and (3) DNA-based identification.

#### 4.3.1. Data Collection Challenges: Spatial Coverage

Marine flora are found in a wide range of habitats, from the intertidal zone to the deep ocean. Conducting a survey that covers all of these habitats can be logistically difficult and time consuming. Considering that French Polynesia consists of some 120 islands with a coastline of 5830 km dispersed over an area of 2,500,000 km^2^, comprehensive sampling covering the whole region is a daunting and unmanageable task. Moreover, some species are locally very limited in geographic distribution (e.g., *Dasya palmatifida* from Afaahiti in Tahiti; *Stypopodium australasicum* from Rapa Island) and, consequently, inconvenient to sample. In our study, we were able to sample a total of 12 islands (c. 10% of all islands). While our study covered intertidal to subtidal areas (down to 60 m depth), sampling was chiefly conducted in shallow depths. We are therefore missing the biodiversity below these depths (e.g., mesophotic depths), which, as illustrated in recent studies, may vary from that of shallower depths (e.g., [131]); also, microhabitats were not sampled. It is evident that we are still missing an important fraction of the total algal diversity from French Polynesia.

#### 4.3.2. Data Collection Challenges: Seasonal Coverage

The abundance and distribution of seaweeds can vary seasonally, depending on factors such as temperature, light, and nutrient availability. Conducting a survey over multiple seasons can help to account for this variability but can also be logistically challenging. Considering that there are two main seasons in French Polynesia (hot and humid from November to March, and drier and cooler from April to October), sampling efforts would need to be doubled, without counting the flora that thrive during inter-seasons. This is especially true for genera of brown algae, for instance, *Colpomenia, Rosenvingea, Hydroclathrus,* and *Sargassum* [14,132]. Many genera (*Hydroclathrus*, *Rosenvingea*, *Pseudochnoospora*) belonging to the family Scytosiphonaceae are blooming seasonally after the cool season between September and December [51]. The abundance of some Dictyotaceae species varies strongly between seasons, such as the genus *Padina,* which has the highest cover during the austral summer [19].

#### 4.3.3. Data Collection Challenges: Taxonomic Expertise

Marine algae are a diverse group of organisms, with many different species that can be difficult to distinguish from one another. Taxonomic expertise is required to accurately identify the different species, which can be time consuming and resource intensive. Marine plant taxonomists usually specialize in specific groups and will develop a sharp eye in the field for their group of expertise. So, while a molecular approach will allow us to uncover hidden diversity, sampling will first and foremost demand from collectors a high discriminatory capacity between taxonomic assemblages in the field, and primary field identification and classification of the collected specimens will require a high level of taxonomic expertise from several specialists for each group to meaningfully synthesize morphological and molecular data [133]. Another critical challenge is that the availability of trained morphological taxonomists is in a worldwide decline due to a lack of recognition and funding for this discipline of research [133].

#### 4.3.4. Data Collection Challenges: Sampling Methods

Marine algae are attached to rocks, the substratum, or other organisms, making it difficult to collect samples without damaging the habitat or the flora itself. Additionally, some epiphytic and turf species are microscopic (e.g., Ceramiales, Rhodophyta), rare, or occur in mixed assemblages or low densities, making them difficult to find, sort, and sample.

#### 4.3.5. Molecular Analyses: DNA Quality and Quantity

Obtaining good-quality DNA from all the species in a marine flora can be challenging, as some species may be difficult to collect, rare, or preserved in a way that is not suitable for DNA extraction. We were able to generate 1007 sequences from 2452 specimens (1823 macroalgae, 619 Cyanobacteria, and 10 seagrasses) from which we extracted DNA. Additionally, some species may have a low DNA yield, which can make barcode amplification difficult.

#### 4.3.6. DNA-Based Identification: Markers and Reference Library

DNA barcoding approaches rely on accurate species identification, and if the reference database contains errors or incomplete information, it can lead to misidentification. As discussed at length in [80], the current challenge with DNA barcoding is the necessity to have a good reference dataset. The reference database is largely lacking in many groups, since not all known species have been sequenced yet, and not all species have been described, with many regions still hosting novelties that remain to be identified. Another major issue is the choice of markers. Currently, no universal marker is used across and within marine floral phyla. For the Rhodophyta and Phaeophyta, *rbc*L is by far the best reference marker, while *tuf*A is preferred for the Chlorophyta [80,111]. Nevertheless, the reference library is far from complete for many taxa within each phylum. For instance, within the Rhodophyta, an *rbc*L library for the Corallinophycidae is largely lacking, and within this group, *cox*1 and *psb*A are preferred markers; similarly, in the Chlorophyta, the *44 tuf*A reference library does not include the Cladophorales, for which nuclear markers are used (e.g., 18S, 28S, ITS). In this sense, [80] highlighted the need to complement the algal GenBank reference libraries, for instance, with mitochondrial markers such as *cox*1 and *cox*3. As a result, there may be difficulties in identifying and classifying all the species in a given marine flora using a single marker per phylum. The marker selection and reference library completeness and quality are likely the main limitations of a standardized approach to DNA barcoding.

#### 4.3.7. DNA-Based Identification: Variation within Species

While DNA barcoding is highly accurate, there can be significant genetic variability within a species, especially across latitudinal gradients, which can make it difficult to distinguish between closely related species in under-sampled regions [134,135]. This can lead to misidentification or the creation of artificial species groups, if the threshold to separate ‘species’ is incorrectly applied. This has led some taxonomists to suggest considering species as ‘discrete evolutionary units’ rather than finite taxa [133].

#### 4.3.8. Cost and Time

Barcoding an entire flora can be a costly and time-consuming process, as each sample must be collected, processed, sequenced, and analyzed. The cost can also vary depending on the sequencing technology used and the number of samples and markers analyzed. We have calculated an average price ranging from USD 50 to 100 per specimen from collection to sequencing.

### 4.4. Utility of DNA Barcoding to Identify Cryptic Species and Ill-Defined Species

DNA barcoding can be especially useful in identifying cryptic and pseudo-cryptic species (i.e., species difficult to distinguish based on morphological characteristics alone) new to science and ill-defined species (e.g., lacking reproductive features to make firm taxonomic decisions). Cryptic species are difficult to find and study in the wild. Through using DNA barcoding, we can identify and distinguish between different species with a high degree of accuracy, even if they have very similar morphological characteristics. Furthermore, DNA barcoding can be used to confirm the identity of species whose names had been previously misapplied.

### 4.5. Conserving the Biodiversity of French Polynesian Marine Flora through DNA Barcoding

As shown in this study, French Polynesia is home to a diverse range of marine plants, including various species of Cyanobacteria, Chlorophyta, Rhodophyta, and Ochrophyta (but only two species of seagrasses). Many of these species are important for ecological reasons, such as providing habitat for other marine organisms, as well as for economic purposes, such as being used for food [136,137,138] and other commercial purposes [139,140]. However, some of these species are threatened by factors such as pollution, coastal development, tourism, and the proliferation of some brown macroalgae (e.g., Dictyotales, Sargassaceae [141]) and Cyanobacteria [20,142]. Marine floral biodiversity is threatened globally, and rapid surveying and monitoring are needed before species are irremediably lost. DNA barcoding can be used to help conserve biodiversity through accurately identifying different species and understanding their genetic relationships [44]. Using DNA barcoding in conjunction with traditional taxonomic expertise and systematic habitat sampling results in a quicker, more precise understanding of species diversity. This knowledge can then be used to inform conservation efforts, such as identifying which species are most in need of protection and which areas are most important for conservation.

In addition to its conservation applications, DNA barcoding can also be used for other purposes, such as detecting the presence of invasive species. The number of alien marine species recorded in the various coastal countries and islands of the world has continued to increase, particularly since the second half of the 20th century, in connection with the increase in maritime traffic and aquaculture exchanges [143]. The morpho-taxonomic methods traditionally employed by many marine monitoring programs are laborious, expensive, and require taxonomic expertise that is often lacking or in decline. In addition, the morpho-taxonomic approach is often unable to detect microscopic invasive species or those at the larval stage, which limits the capacity for eradication, especially at the first stage of introduction. The recent development of molecular tools (e.g., metabarcoding), in particular high-throughput DNA sequencing such as environmental DNA monitoring [143], offers enormous advantages in marine monitoring due to its dual capacity to detect invasive species at any stage of development, while producing a comprehensive and holistic view of all biological communities present in any type of environmental sample (water, sediment, biofilm/biofouling [144,145]). However, it implies that reference molecular data are available, and its effectiveness depends on the choice of genetic markers used. These techniques are particularly effective at detecting targeted species, including when they are present in low density, and make it possible to massively increase the number of sites studied. They are thus complementary to traditional techniques [146]. A survey based on eDNA metabarcoding would have revealed 22 species of introduced seaweeds—of which only one species was confirmed with barcoding, *Solieria filiformis*)—including 21 Rhodophyta and 1 Ochrophyta (*Colpomenia sinuosa*) [147]. The latter study defined “introduced” as “non-native” based on the information available on the database WORMS [148] (i.e., to determine a species geographic origin). Nevertheless, comprehensive large-scale phylogeographic analyses are necessary to conclusively determine the natural geographic range of a given species [149]. For instance, [148] listed the cosmopolitan species *C. sinuosa* as an introduced species in French Polynesia. However, a global phylogeographic study conducted on this taxon showed that it consisted of a species complex with high genetic diversity mainly associated with its geographic distribution [150]. In our study, *C. sinuosa* belonged to two groups within the *C. sinuosa* complex, both broadly distributed in the Pacific, thus not supporting the hypothesis of an introduction. On the other hand, another species, *Gracilaria caudata*, with a clear native range in the Atlantic [151], was identified from Tahiti in our study—but not in [147].

## 5. Conclusions

This study demonstrated how DNA barcoding is a powerful tool that can help to document and conserve the biodiversity of marine flora worldwide. DNA barcoding can help to create a more accurate picture of biodiversity and inform conservation efforts to protect local species. Our revised listing of French Polynesian marine flora now contains a total of 670 species, including 2 species of Alismatales, 146 Cyanobacteria, 315 Rhodophyta, 92 Ochrophyta, and 115 Chlorophyta, a nearly two-fold increase from previous estimates. Further systematic studies (both molecular and using traditional taxonomic expertise) will be needed to validate the taxonomic identity of the numerous new molecular lineages reported in this study. The DNA database presented in this study has the potential to serve as a valuable reference library for identification purposes, making a significant contribution to the advancement of molecular taxonomy, ecological research, and biodiversity studies in French Polynesia.

## Figures and Tables

**Figure 1 biology-12-01124-f001:**
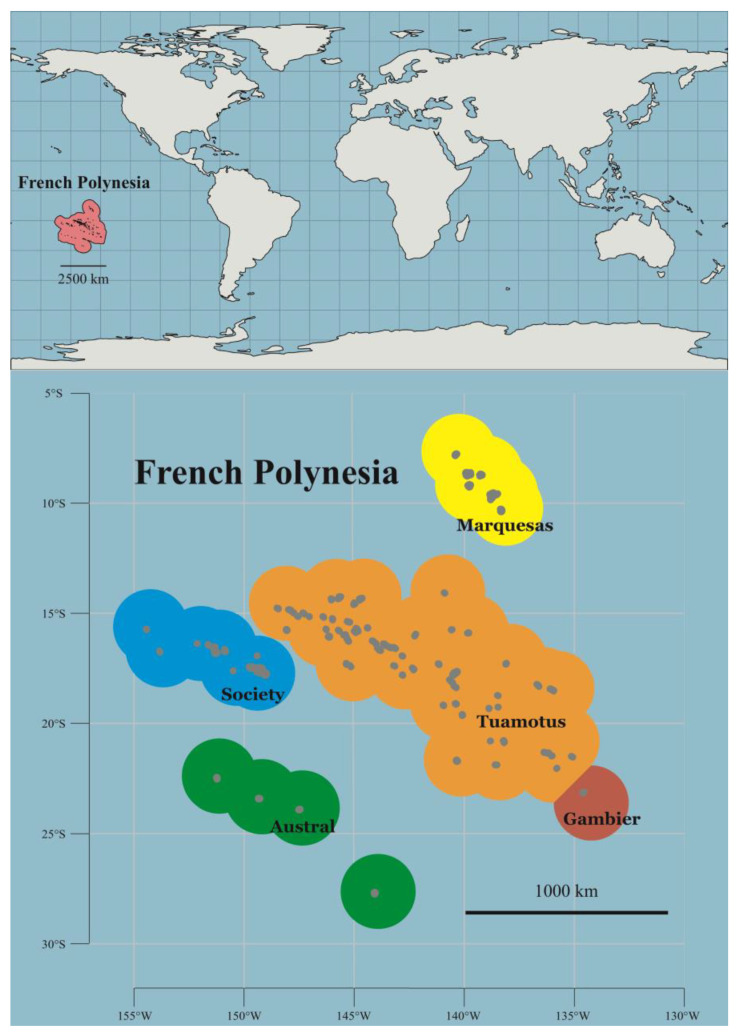
Map of French Polynesia showing the five main archipelagos.

**Figure 2 biology-12-01124-f002:**
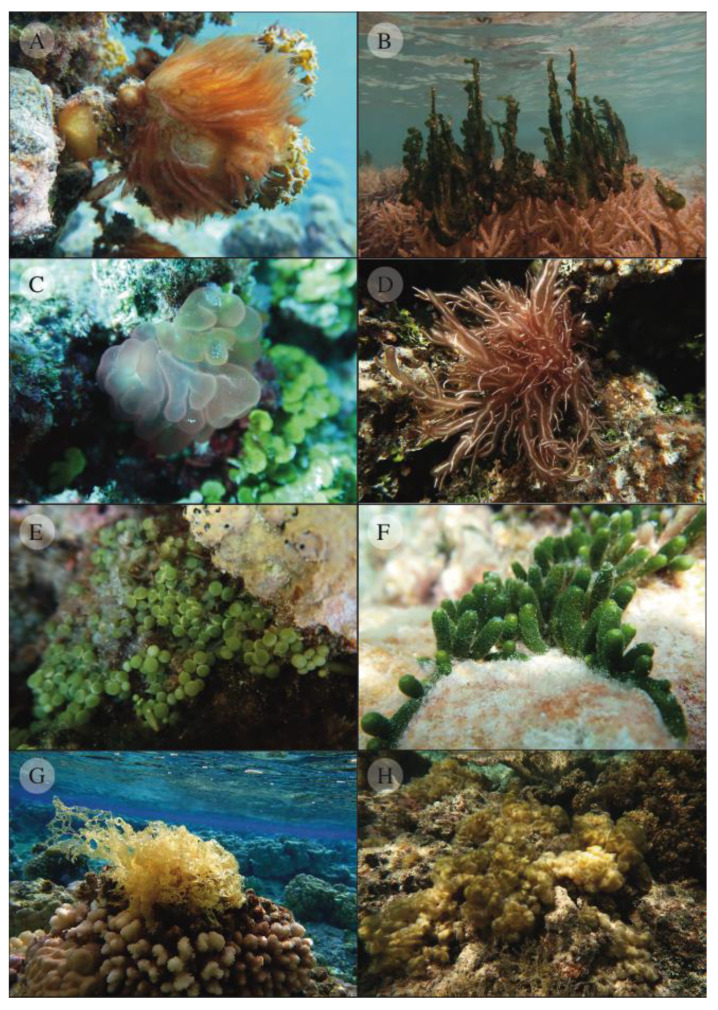
Illustrations of some marine flora from French Polynesia. Two cyanobacteria—*Caldora pennicilata* (**A**), *Anabaena* sp.1 (**B**), and some macroalgae—*Gibsmithia dotyi* (**C**), *Trichogloea requienii* (**D**), *Caulerpa chemnitzia* (**E**), *Caulerpa webbiana* var. *pickeringii* (**F**), *Manzaea minuta* (**G**), *Colpomenia claytoniae* (**H**).

**Table 1 biology-12-01124-t001:** Catalog of marine flora (Alismatales, Chlorophyta, Cyanobacteria, Ochrophyta, Rhodophyta) from French Polynesia based on molecular and morphological identifications and compiled from published past records and the present study.

Order	Family	Species Name	Molecular Confirmation	Endemic	Morphology	Regional DNA Sequence	References	Markers Sequenced
Acrochaetiales	Acrochaetiaceae	*Acrochaetium barbadense* (Vickers) Børgesen	N.S.	N	Y	N	[16]	
Acrochaetiales	Acrochaetiaceae	*Acrochaetium microscopicum* (Nägeli ex Kützing) Nägeli	N.S.	N	Y	N	[16]	
Bonnemaisoniales	Bonnemaisoniaceae	*Asparagopsis taxiformis* (Delile) Trevisan	CONF.	N	Y	Y	This study, [16,56]	*cox*1, *rbc*L, 28S
Ceramiales	Callithamniaceae	*Aglaothamnion boergesenii* (Aponte & D.L.Ballantine) L’Hardy-Halos & Rueness	N.S.	N	Y	N	[77]	
Ceramiales	Callithamniaceae	*Aglaothamnion* sp.1FP	N.L.	Y	Y	Y	This study	*rbc*L
Ceramiales	Callithamniaceae	*Aglaothamnion* sp.2FP	N.L.	Y	Y	Y	This study	*rbc*L
Ceramiales	Callithamniaceae	*Crouania attenuata* (C.Agardh) J.Agardh	CONF.	N	Y	Y	[16]	*rbc*L
Ceramiales	Callithamniaceae	*Seirospora orientalis* Kraft	N.S.	N	Y	N	[16]	
Ceramiales	Callithamniaceae	*Spyridia americana* Durant	N.L.	N	Y	Y	This study	*rbc*L
Ceramiales	Callithamniaceae	*Spyridia filamentosa* (Wulfen) Harvey	N.S.	N	Y	N	[16]	
Ceramiales	Callithamniaceae	*Spyridia hypnoides* (Bory) Papenfuss	N.S.	N	Y	N	[16]	
Ceramiales	Callithamniaceae	*Spyridia* sp.1FP	N.L.	Y	Y	Y	This study	*rbc*L
Ceramiales	Callithamniaceae	*Spyridia* sp.2FP	N.L.	Y	Y	Y	This study	*rbc*L
Ceramiales	Ceramiaceae	*Acrothamnion butlerae* (Collins) Kylin	N.S.	N	Y	N	[77]	
Ceramiales	Ceramiaceae	*Antithamnion antillanum* Børgesen	N.S.	N	Y	N	[16]	
Ceramiales	Ceramiaceae	*Antithamnion decipiens* (J.Agardh) Athanasiadis	N.S.	N	Y	N	[16]	
Ceramiales	Ceramiaceae	*Antithamnionella breviramosa* (E.Y.Dawson) Wollaston	N.S.	N	Y	N	[16]	
Ceramiales	Ceramiaceae	*Antithamnionella elegans* (Berthold) J.H.Price & D.M.John	N.S.	N	Y	Y	[77]	
Ceramiales	Ceramiaceae	*Centroceras clavulatum* (C.Agardh) Montagne	N.S.	N	Y	N	[16]	
Ceramiales	Ceramiaceae	*Centroceras minutum* Yamada	N.S.	N	Y	N	[16]	
Ceramiales	Ceramiaceae	*Centroceras* sp.1FP (cf. *Centroceras gasparrinii*)	N.L.	Y	Y	Y	This study	*rbc*L
Ceramiales	Ceramiaceae	*Centroceras* sp.2FP	N.L.	Y	Y	Y	This study	*rbc*L
Ceramiales	Ceramiaceae	*Centroceras* sp.3FP	N.L.	Y	Y	Y	This study	*rbc*L
Ceramiales	Ceramiaceae	*Centroceras* sp.4FP	N.L.	Y	Y	Y	This study	*rbc*L
Ceramiales	Ceramiaceae	*Centroceras* sp.5FP	N.L.	Y	Y	Y	This study	*rbc*L
Ceramiales	Ceramiaceae	*Ceramium aduncum* Nakamura	N.S.	N	Y	N	[16]	
Ceramiales	Ceramiaceae	*Ceramium borneense* Weber Bosse	N.S.	N	Y	N	[16]	
Ceramiales	Ceramiaceae	*Ceramium codii* (H.Richards) Mazoyer	N.S.	N	Y	N	[16]	
Ceramiales	Ceramiaceae	*Ceramium* sp.1FP	N.L.	Y	Y	Y	This study	*rbc*L
Ceramiales	Ceramiaceae	*Ceramium* sp.2FP	N.L.	Y	Y	Y	This study	*rbc*L
Ceramiales	Ceramiaceae	*Ceramium* sp.3FP	N.L.	Y	Y	Y	This study	*rbc*L
Ceramiales	Ceramiaceae	*Ceramium* sp.4FP	N.L.	Y	Y	Y	This study	*rbc*L
Ceramiales	Ceramiaceae	*Ceramium* sp.5FP	N.L.	Y	Y	Y	This study	*rbc*L
Ceramiales	Ceramiaceae	*Ceramium upolense* G.R.South & Skelton	N.S.	N	Y	N	[16]	
Ceramiales	Ceramiaceae	*Ceramium vagans* P.C.Silva	N.S.	N	Y	N	[16]	
Ceramiales	Ceramiaceae	*Corallophila kleiwegii* Weber Bosse	N.S.	N	Y	N	[16]	
Ceramiales	Ceramiaceae	*Gayliella macrotricha* (Feldmann-Mazoyer) Huisman	N.S.	N	Y	N	[16]	
Ceramiales	Ceramiaceae	*Gayliella* sp.1 FP	N.L.	Y	Y	Y	This study	*rbc*L
Ceramiales	Ceramiaceae	*Gayliella* sp.2 FP	N.L.	Y	Y	Y	This study	*rbc*L
Ceramiales	Ceramiaceae	*Gayliella* sp.3 FP	N.L.	Y	Y	Y	This study	*rbc*L
Ceramiales	Ceramiaceae	*Gayliella* sp.4 FP	N.L.	Y	Y	Y	This study	*rbc*L
Ceramiales	Ceramiaceae	*Gayliella* sp.5 FP	N.L.	Y	Y	Y	This study	*rbc*L
Ceramiales	Ceramiaceae	*Gayliella transversalis* (Collins & Hervey) T.O.Cho & Fredericq	N.S.	N	Y	N	[16]	
Ceramiales	Delesseriaceae	*Dasya anastomosans* (Weber Bosse) M.J.Wynne	CONF.	N	Y	Y	This study, [16]	*rbc*L
Ceramiales	Delesseriaceae	*Dasya iyengarii* Børgesen	N.S.	N	Y	N	[16]	
Ceramiales	Delesseriaceae	*Dasya mollis* Harvey	N.S.	N	Y	N	[16]	
Ceramiales	Delesseriaceae	*Dasya murrayana* I.A.Abbott & A.J.K.Millar	N.S.	N	Y	N	[16]	
Ceramiales	Delesseriaceae	*Dasya palmatifida* (Weber Bosse) A.J.K.Millar & Coppejans	N.S.	N	Y	N	[16]	
Ceramiales	Delesseriaceae	*Dasya pedicellata* (C.Agardh) C.Agardh	N.S.	N	Y	N	[16]	
Ceramiales	Delesseriaceae	*Dasya* sp.1FP	N.L.	Y	Y	Y	This study	*rbc*L
Ceramiales	Delesseriaceae	*Dasya* sp.2FP	N.L.	Y	Y	Y	This study	*rbc*L
Ceramiales	Delesseriaceae	*Dasya* sp.3FP	N.L.	Y	Y	Y	This study	*rbc*L
Ceramiales	Delesseriaceae	*Hypoglossum simulans* M.J.Wynne, I.R.Price & D.L.Ballantine	cf. Hypoglossum sp.1	N	Y	N	[16]	
Ceramiales	Delesseriaceae	*Hypoglossum* sp.1FP	N.L.	Y	Y	Y	This study	*rbc*L
Ceramiales	Delesseriaceae	*Martensia fragilis* Harvey	N.S.	N	Y	N	[16]	
Ceramiales	Delesseriaceae	*Myriogramme cartilaginea* (Harvey) Womersley	N.S.	N	Y	N	[16]	
Ceramiales	Delesseriaceae	*Nitophyllum adhaerens* M.J.Wynne	N.S.	N	Y	N	[16]	
Ceramiales	Neuroglosseae	*Schizoseris* sp.1FP	N.L.	Y	Y	Y	This study	*rbc*L
Ceramiales	Rhodomelaceae	*Acanthophora pacifica* (Setchell) Kraft	N.S.	N	Y	Y	This study, [16]	
Ceramiales	Rhodomelaceae	*Acanthophora spicifera* (M.Vahl) Børgesen	CONF.	N	Y	Y	This study, [16]	*rbc*L
Ceramiales	Rhodomelaceae	*Amansia rhodantha* (Harvey) J.Agardh	N.L.	N	Y	Y	This study, [16]	*rbc*L
Ceramiales	Rhodomelaceae	*Bostrychia moritzian* (Sonder ex Kützing) J.Agardh	N.S.	N	Y	N	[16]	
Ceramiales	Rhodomelaceae	*Bostrychia tenella* (J.V.Lamouroux) J.Agardh	N.S.	N	Y	N	[16]	
Ceramiales	Rhodomelaceae	*Chondria arcuata* Hollenberg	N.S.	N	Y	N	[16]	
Ceramiales	Rhodomelaceae	*Chondria bullata* N’Yeurt & Payri	N.S.	N	Y	N	[16]	
Ceramiales	Rhodomelaceae	*Chondria dangeardii* E.Y.Dawson	N.S.	N	Y	N	[16]	
Ceramiales	Rhodomelaceae	*Chondria dasyphylla* (Woodward) C.Agardh	N.S.	N	Y	N	[16]	
Ceramiales	Rhodomelaceae	*Chondria minutula* Weber Bosse	N.S.	N	Y	N	[16]	
Ceramiales	Rhodomelaceae	*Chondria repens* Børgesen	N.S.	N	Y	N	[16]	
Ceramiales	Rhodomelaceae	*Chondria simpliciuscula* Weber Bosse	N.S.	N	Y	N	[16]	
Ceramiales	Rhodomelaceae	*Chondria* sp.1FP	N.L.	Y	Y	Y	This study	*rbc*L
Ceramiales	Rhodomelaceae	*Chondria* sp.2FP	N.L.	Y	Y	Y	This study	*rbc*L
Ceramiales	Rhodomelaceae	*Chondria* sp.3FP	N.L.	Y	Y	Y	This study	*rbc*L
Ceramiales	Rhodomelaceae	*Chondria* sp.4FP	N.L.	Y	Y	Y	This study	*rbc*L
Ceramiales	Rhodomelaceae	*Chondria* sp.5FP	N.L.	Y	Y	Y	This study	*rbc*L
Ceramiales	Rhodomelaceae	*Chondria* sp.6FP	N.L.	Y	Y	Y	This study	*rbc*L
Ceramiales	Rhodomelaceae	*Chondria* sp.7FP	N.L.	Y	Y	Y	This study	*rbc*L
Ceramiales	Rhodomelaceae	*Chondrophycus* sp.1FP	N.L.	Y	Y	Y	This study	*rbc*L
Ceramiales	Rhodomelaceae	*Chondrophycus* sp.2FP	N.L.	Y	Y	Y	This study	*rbc*L
Ceramiales	Rhodomelaceae	*Chondrophycus* sp.3FP	N.L.	Y	Y	Y	This study	*rbc*L
Ceramiales	Rhodomelaceae	*Chondrophycus succisus* (A.B.Cribb) K.W.Nam	N.S.	N	Y	N	[16]	
Ceramiales	Rhodomelaceae	*Ditria reptans* Hollenberg	N.S.	N	Y	N	[16]	
Ceramiales	Rhodomelaceae	*Herposiphonia delicatula* Hollenberg	N.S.	N	Y	N	[16]	
Ceramiales	Rhodomelaceae	*Herposiphonia dendroidea* Hollenberg	N.S.	N	Y	N	[16]	
Ceramiales	Rhodomelaceae	*Herposiphonia pacifica* Hollenberg	N.S.	N	Y	N	[16]	
Ceramiales	Rhodomelaceae	*Herposiphonia parca* Setchell	N.S.	N	Y	N	[16]	
Ceramiales	Rhodomelaceae	*Herposiphonia secunda* (C.Agardh) Ambronn	N.S.	N	Y	N	[16]	
Ceramiales	Rhodomelaceae	*Herposiphonia* sp.1FP	N.L.	Y	Y	Y	This study	*rbc*L
Ceramiales	Rhodomelaceae	*Herposiphonia* sp.2FP	N.L.	Y	Y	Y	This study	*rbc*L
Ceramiales	Rhodomelaceae	*Herposiphonia* sp.3FP	N.L.	Y	Y	Y	This study	*rbc*L
Ceramiales	Rhodomelaceae	*Heterosiphonia crispella* var. *laxa* (Børgesen) M.J.Wynne	N.S.	N	Y	N	[16]	
Ceramiales	Rhodomelaceae	*Heterosiphonia gibbesii* (Harvey) Falkenberg	N.S.	N	Y	N	[16]	
Ceramiales	Rhodomelaceae	*Laurencia caraibica* P.C.Silva	N.S.	N	Y	N	[16]	
Ceramiales	Rhodomelaceae	*Laurencia claviformis* Børgesen	N.S.	N	Y	N	[16]	
Ceramiales	Rhodomelaceae	*Laurencia decumbens* Kützing	N.S.	N	Y	N	[16]	
Ceramiales	Rhodomelaceae	*Laurencia glandulifera* (Kützing) Kützing	N.S.	N	Y	N	[16]	
Ceramiales	Rhodomelaceae	*Laurencia* sp.1FP	N.L.	Y	Y	Y	This study	*rbc*L
Ceramiales	Rhodomelaceae	*Laurencia* sp.2FP	N.L.	Y	Y	Y	This study	*rbc*L
Ceramiales	Rhodomelaceae	*Laurencia* sp.3FP	N.L.	Y	Y	Y	This study	*rbc*L
Ceramiales	Rhodomelaceae	*Laurencia* sp.4FP	N.L.	Y	Y	Y	This study	*rbc*L
Ceramiales	Rhodomelaceae	*Melanothamnus apiculatus* (Hollenberg) Díaz- Tapia & Maggs	N.S.	N	Y	N	[16]	
Ceramiales	Rhodomelaceae	*Melanothamnus delicatulus* (Hollenberg) Huisman	N.S.	N	Y	N	[16]	
Ceramiales	Rhodomelaceae	*Melanothamnus ecorticatus* (R.E.Norris) Díaz-Tapia & Maggs	N.S.	N	Y	N	[16]	
Ceramiales	Rhodomelaceae	*Melanothamnus ferulaceus* (Suhr ex J.Agardh) Díaz-Tapia & Maggs	N.S.	N	Y	N	[16]	
Ceramiales	Rhodomelaceae	*Melanothamnus savatieri* (Hariot) Díaz-Tapia & Maggs	N.S.	N	Y	N	[16]	
Ceramiales	Rhodomelaceae	*Melanothamnus sphaerocarpus* (Børgesen) Díaz-Tapia & Maggs	N.S.	N	Y	N	[16]	
Ceramiales	Rhodomelaceae	*Melanothamnus tongatensis* (Harvey ex Kützing) Díaz-Tapia & Maggs	N.S.	N	Y	N	[16]	
Ceramiales	Rhodomelaceae	*Melanothamnus upolensis* (Grunow) Díaz-Tapia & Maggs	N.S.	N	Y	N	[16]	
Ceramiales	Rhodomelaceae	*Ohelopapa flexilis* (Setchell) F.Rousseau, Martin-Lescanne, Payri & L.Le Gall	CONF.	N	Y	Y	This study, [16,63]	*rbc*L
Ceramiales	Rhodomelaceae	*Palisada cervicornis* (Harvey) Collado-Vides, Cassano & M.T.Fujii	N.S.	N	Y	N	[16]	
Ceramiales	Rhodomelaceae	*Palisada crustiformans* (K.J.McDermid) A.R.Sherwood, A.Kurihara & K.W.Nam	N.S.	N	Y	N	[16]	
Ceramiales	Rhodomelaceae	*Palisada parvipapillata* (C.K.Tseng) K.W.Nam	N.S.	N	Y	N	[16]	
Ceramiales	Rhodomelaceae	*Palisada perforata* (Bory) K.W.Nam	N.S.	N	Y	N	[16]	
Ceramiales	Rhodomelaceae	*Palisada* sp.1FP	N.L.	Y	Y	Y	This study	*rbc*L
Ceramiales	Rhodomelaceae	*Palisada* sp.2FP	N.L.	Y	Y	Y	This study	*rbc*L
Ceramiales	Rhodomelaceae	*Palisada* sp.3FP	N.L.	Y	Y	Y	This study	*rbc*L
Ceramiales	Rhodomelaceae	*Palisada yamadana* (M.Howe) K.W.Nam	N.S.	N	Y	N	[16]	
Ceramiales	Rhodomelaceae	*Phaeocolax kajimurae* Hollenberg	N.S.	N	Y	N	[16]	
Ceramiales	Rhodomelaceae	*Polysiphonia dotyi* Hollenberg	N.S.	N	Y	N	[16]	
Ceramiales	Rhodomelaceae	*Polysiphonia homoia* Setchell & N.L.Gardner	N.S.	N	Y	N	[16]	
Ceramiales	Rhodomelaceae	*Polysiphonia poko* Hollenberg	N.S.	N	Y	N	[16]	
Ceramiales	Rhodomelaceae	*Polysiphonia scopulorum* Harvey	N.S.	N	Y	N	[16]	
Ceramiales	Rhodomelaceae	*Polysiphonia* sp.1FP	N.L.	Y	Y	Y	This study	*rbc*L
Ceramiales	Rhodomelaceae	*Polysiphonia* sp.2FP	N.L.	Y	Y	Y	This study	*rbc*L
Ceramiales	Rhodomelaceae	*Polysiphonia* sp.3FP	N.L.	Y	Y	Y	This study	*rbc*L
Ceramiales	Rhodomelaceae	*Polysiphonia* sp.4FP	N.L.	Y	Y	Y	This study	*rbc*L
Ceramiales	Rhodomelaceae	*Polysiphonia* sp.5FP *Polysiphonia sertularioides* complex	N.L.	Y	Y	Y	This study	*rbc*L
Ceramiales	Rhodomelaceae	*Spirocladia barodensis* Børgesen	N.S.	N	Y	N	[16]	
Ceramiales	Rhodomelaceae	*Womersleyella herpa* (Hollenberg) R.E.Norris	N.S.	N	Y	N	[16]	
Ceramiales	Rhodomelaceae	*Womersleyella setacea* (Hollenberg) R.E.Norris	N.S.	N	Y	N	[16]	
Ceramiales	Wrangeliaceae	*Anotrichium* sp.1FP	N.L.	Y	Y	Y	This study	*rbcL*
Ceramiales	Wrangeliaceae	*Anotrichium* sp.2FP	N.L.	Y	Y	Y	This study	*rbcL*
Ceramiales	Wrangeliaceae	*Anotrichium tenue* (C.Agardh) Nägeli	N.S.	N	Y	N	[16]	
Ceramiales	Wrangeliaceae	*Griffithsia schousboei* Montagne	N.S.	N	Y	N	[16]	
Ceramiales	Wrangeliaceae	*Griffithsia* sp.1FP	N.L.	Y	Y	Y	This study	*rbc*L
Ceramiales	Wrangeliaceae	*Griffithsia* sp.2FP	N.L.	Y	Y	Y	This study	*rbc*L
Ceramiales	Wrangeliaceae	*Griffithsia* sp.3FP	N.L.	Y	Y	Y	This study	*rbc*L
Ceramiales	Wrangeliaceae	*Griffithsia* sp.4FP	N.L.	Y	Y	Y	This study	*rbc*L
Ceramiales	Wrangeliaceae	*Haloplegma duperreyi* Montagne	N.S.	N	Y	N	[16]	
Ceramiales	Wrangeliaceae	*Haloplegma* sp.1FP	N.L.	Y	Y	Y	This study	*rbc*L
Ceramiales	Wrangeliaceae	*Ptilothamnion cladophorae* (Yamada & T.Tanaka) G.Feldmann-Mazoyer	N.S.	N	Y	N	[16]	
Ceramiales	Wrangeliaceae	*Wrangelia* sp.1FP	N.L.	Y	Y	Y	This study	*rbc*L
Corallinales	Corallinaceae	*Dawsoniolithon conicum* (E.Y.Dawson) Caragnano, Foetisch, Maneveldt & Payri	CONF.	N	Y	Y	[16,59]	*cox*1, *psb*A, SSU, LSU
Corallinales	Corallinaceae	*Ellisolandia elongata* (J.Ellis & Solander) K.R.Hind & G.W.Saunders	N.S.	N	Y	N	[16]	
Corallinales	Corallinaceae	*Jania acutiloba* (Decaisne) J.H.Kim, Guiry & H.-G.Choi	cf. Jania sp.1	N	Y	N	[16]	
Corallinales	Corallinaceae	*Jania articulata* N’Yeurt & Payri	N.S.	N	Y	N	[16]	
Corallinales	Corallinaceae	*Jania pedunculata* var. *adhaerens* (J.V.Lamouroux) A.S.Harvey, Woelkerling & Reviers	N.S.	N	Y	N	[16]	
Corallinales	Corallinaceae	*Jania pumila* J.V.Lamouroux	N.S.	N	Y	N	[16]	
Corallinales	Corallinaceae	*Jania rubens* (Linnaeus) J.V.Lamouroux	N.S.	N	Y	N	[16]	
Corallinales	Corallinaceae	*Jania* sp.1FP	N.L.	Y	Y	Y	This study	*rbc*L
Corallinales	Corallinaceae	*Jania* sp.2FP	N.L.	Y	Y	Y	This study	*rbc*L
Corallinales	Corallinaceae	*Jania spectabilis* (Harvey ex Grunow) J.H.Kim, Guiry & H.-G.Choi	N.S.	N	Y	N	[16]	
Corallinales	Corallinaceae	*Jania subulata* (Ellis & Solander) Sonder	N.S.	N	Y	N	[16]	
Corallinales	Corallinaceae	*Parvicellularium* sp.1FP	CONF.	Y	?	Y	[59]	*psb*A, LSU
Corallinales	Hydrolithaceae	*Hydrolithon boergesenii* (Foslie) Foslie	N.S.	N	Y	N	[16]	
Corallinales	Hydrolithaceae	*Hydrolithon boreale* (Foslie) Y.M.Chamberlain	N.S.	N	Y	N	[16]	
Corallinales	Hydrolithaceae	*Hydrolithon farinosum* (J.V.Lamouroux) Penrose & Y.M.Chamberlain	N.S.	N	Y	N	[16]	
Corallinales	Hydrolithaceae	*Hydrolithon murakoshii* Iryu & Matsuda	N.S.	N	Y	N	[16]	
Corallinales	Lithophyllaceae	*Amphiroa anceps* (Lamarck) Decaisne	N.S.	N	Y	N	[16]	
Corallinales	Lithophyllaceae	*Amphiroa foliacea* J.V.Lamouroux	N.S.	N	Y	N	[16]	
Corallinales	Lithophyllaceae	*Amphiroa* sp.1FP	N.L.	Y	Y	Y	This study	*rbc*L
Corallinales	Lithophyllaceae	*Amphiroa* sp.2FP	N.L.	Y	Y	Y	This study	*rbc*L
Corallinales	Lithophyllaceae	*Amphiroa* sp.3FP	N.L.	Y	Y	Y	This study	*rbc*L
Corallinales	Lithophyllaceae	*Amphiroa valonioides* Yendo	N.L.	N	Y	Y	This study, [16]	*rbc*L
Corallinales	Lithophyllaceae	*Lithophyllum* sp.1FP	YES: [60]	Y	Y	Y	[60]	*rbc*L
Corallinales	Lithophyllaceae	*Lithophyllum flavescens* Keats	N.S.	N	Y	N	[16]	
Corallinales	Lithophyllaceae	*Lithophyllum insipidum* W.H.Adey, R.A.Townsend & Boykins	N.S.	N	Y	N	[16]	
Corallinales	Lithophyllaceae	*Lithophyllum kotschyanum* Unger	N.S.	N	Y	N	[16]	
Corallinales	Lithophyllaceae	*Titanoderma pustulatum* (J.V.Lamouroux) Nägeli	N.S.	N	Y	N	[16]	
Corallinales	Mastophoraceae	*Mastophora pacifica* (Heydrich) Foslie	NOT CONF.	N	Y	N	[16]	*rbc*L
Corallinales	Mesophyllumaceae	*Mesophyllum erubescens* (Foslie) Me.Lemoine	N.S.	N	Y	N	[16]	
Corallinales	Mesophyllumaceae	*Mesophyllum funafutiense* (Foslie) Verheij	N.S.	N	Y	N	[16]	
Corallinales	Porolithaceae	*Harveylithon rupestre* (Foslie) A.Rösler, Perfectti, V.Peña & J.C.Braga	N.S.	N	Y	N	[16]	
Corallinales	Porolithaceae	*Harveylithon samoënse* (Foslie) A.Rösler, Perfectti, V.Peña & J.C.Braga	N.S.	N	Y	N	[16]	
Corallinales	Porolithaceae	*Porolithon gardineri* (Foslie) Foslie	N.S.	N	Y	N	[16]	
Corallinales	Porolithaceae	*Porolithon onkodes* (Heydrich) Foslie	N.S.	N	Y	N	[16]	
Corallinales	Porolithaceae	*Porolithon* sp.1FP	CONF.	Y	?	Y	[59]	*cox*1, *psb*A, SSU, LSU
Corallinales	Porolithaceae	*Porolithon* sp.2FP	CONF.	Y	?	Y	[59]	*cox*1, *psb*A, SSU, LSU
Corallinales	Spongitidaceae	*Neogoniolithon brassica-florida* (Harvey) Setchell & L.R.Mason	CONF.	N	Y	Y	[16,61]	28S
Corallinales	Spongitidaceae	*Neogoniolithon fosliei* (Heydrich) Setchell & L.R.Mason	CONF.	N	Y	Y	[16,61]	18S, 23S
Corallinales	Spongitidaceae	*Neogoniolithon frutescens* (Foslie) Setchell & L.R.Mason	CONF.	N	?	Y	[61,62]	*cox*1, *psb*A, *rbc*L, 28S
Corallinales	Spongitidaceae	*Neogoniolithon megalocystum* (Foslie) Setchell & L.R.Mason	N.S.	N	Y	N	[16]	
Corallinales	Spongitidaceae	*Spongites* sp.1FP	CONF.	Y	?	Y	[59]	*cox*1, *psb*A, *rbc*L, 28S
Erythropeltales	Erythrotrichiaceae	*Erythrotrichia carnea* (Dillwyn) J.Agardh	N.S.	N	Y	N	[16]	
Gelidiales	Gelidiaceae	*Gelidium isabelae* W.R.Taylor	N.S.	N	Y	N	[16]	
Gelidiales	Gelidiaceae	*Gelidium samoense* Reinbold	N.S.	N	Y	N	[16]	
Gelidiales	Gelidiellaceae	*Gelidiella acerosa* (Forsskål) Feldmann & Hamel	CONF.	N	Y	Y	This study, [16]	*rbc*L
Gelidiales	Gelidiellaceae	*Gelidiella damseliana* Huisman, G.H.Boo & S.M.Boo	N.L.	N	Y	Y	This study	*rbc*L
Gelidiales	Gelidiellaceae	*Gelidiella fanii* S.- M.Lin	N.R.	N	Y	Y	This study, [54]	*cox*1, *rbc*L
Gelidiales	Gelidiellaceae	*Gelidiella machrisiana* E.Y.Dawson	N.S.	N	Y	N	[16]	
Gelidiales	Gelidiellaceae	*Parviphycus antipae* (Celan) B.Santelices	N.S.	N	Y	N	[16]	
Gelidiales	Orthogonacladiaceae	*Aphanta ligulata* Huisman, G.H.Boo & S.M.Boo	N.L.	N	Y	Y	This study	*rbc*L
Gelidiales	Pterocladiaceae	*Pterocladiella* sp.1FP	N.L.	Y	Y	Y	This study	*rbc*L
Gelidiales	Pterocladiaceae	*Pterocladiella* sp.2FP	N.L.	Y	Y	Y	This study	*rbc*L
Gelidiales	Pterocladiaceae	*Pterocladiella* sp.3FP	N.L.	Y	Y	Y	This study	*rbc*L
Gelidiales	Pterocladiaceae	*Pterocladiella caerulescens* (Kützing) Santelices & Hommersand	N.S.	N	Y	N	[16]	
Gelidiales	Pterocladiaceae	*Pterocladiella caloglossoides* (M.Howe) Santelices	N.S.	N	Y	N	[16]	
Gigartinales	Caulacanthaceae	*Caulacanthus ustulatus* (Turner) Kützing	N.S.	N	Y	N	[16]	
Gigartinales	Chondrymeniaceae	*Dissimularia tauensis* G.T.Kraft & G.W.Saunders	N.L.	N	Y	Y	This study	*rbc*L
Gigartinales	Chondrymeniaceae	*Dissimularia umbraticola* (E.Y.Dawson) G.T.Kraft & G.W.Saunders	CONF.	N	Y	Y	This study, [16]	*rbc*L
Gigartinales	Corynocystaceae	*Corynocystis prostrata* Kraft	N.L.	N	Y	Y	This study, [16]	*rbc*L
Gigartinales	Cystocloniaceae	*Calliblepharis saidana* (Holmes) M.Y.Yang & M.S.Kim	N.S.	N	Y	N	[16]	
Gigartinales	Cystocloniaceae	*Hypnea esperi* Bory, nom. illeg.	N.S.	N	Y	N	[16]	
Gigartinales	Cystocloniaceae	*Hypnea pannosa* J.Agardh	CONF.	N	Y	Y	This study, [16]	*rbc*L
Gigartinales	Cystocloniaceae	*Hypnea* sp.1FP	N.L.	?	Y	Y	This study	*rbc*L
Gigartinales	Cystocloniaceae	*Hypnea* sp.2FP	N.L.	?	Y	Y	This study	*rbc*L
Gigartinales	Cystocloniaceae	*Hypnea* sp.3FP	N.L.	?	Y	Y	This study	*rbc*L
Gigartinales	Cystocloniaceae	*Hypnea* sp.4FP	N.L.	?	Y	Y	This study	*rbc*L
Gigartinales	Cystocloniaceae	*Hypnea* sp.5FP	N.L.	?	Y	Y	This study	*rbc*L
Gigartinales	Cystocloniaceae	*Hypnea* sp.6FP	N.L.	?	Y	Y	This study	*rbc*L
Gigartinales	Cystocloniaceae	*Hypnea* sp.7FP	N.L.	?	Y	Y	This study	*rbc*L
Gigartinales	Cystocloniaceae	*Hypnea* sp.8FP	N.L.	?	Y	Y	This study	*rbc*L
Gigartinales	Cystocloniaceae	*Hypnea* sp.9FP	N.L.	?	Y	Y	This study	*rbc*L
Gigartinales	Cystocloniaceae	*Hypnea spinella* (C.Agardh) Kützing	N.S.	N	Y	N	[16]	
Gigartinales	Dicranemataceae	*Tylotus* sp.1FP	N.L.	?	Y	Y	This study	*rbc*L
Gigartinales	Dumontiaceae	*Dudresnaya hawaiiensis* R.K.S.Lee	N.S.	N	Y	N	[16]	
Gigartinales	Dumontiaceae	*Gibsmithia dotyi* Kraft & R.W.Ricker	CONF.	N	Y	Y	[77]; This study	*rbc*L
Gigartinales	Dumontiaceae	*Gibsmithia hawaiiensis* Doty	N.S.	N	Y	Y	[16]	
Gigartinales	Dumontiaceae	*Gibsmithia indopacifica* D.Gabriel, Draisma & Fredericq	N.R.	N	Y	Y	This study, [55]	*cox*1, *rbc*L, 23S
Gigartinales	Dumontiaceae	*Gibsmithia larkumii* Kraft	N.S.	N	Y	N	[16]	
Gigartinales	Dumontiaceae	*Gibsmithia* sp.1FP	N.L.	Y	Y	Y	This study	*rbc*L
Gigartinales	Gloiosiphoniaceae	*Peleophycus multiprocarpius* I.A.Abbott	N.S.	N	Y	N	[16]	
Gigartinales	Kallymeniaceae	*Kallymenia thompsonii* I.A.Abbott & McDermid	N.S.	N	Y	N	[16]	
Gigartinales	Kallymeniaceae	*Meredithia* sp.1FP	N.L.	Y	Y	Y	This study	*rbc*L
Gigartinales	Phyllophoraceae	*Ahnfeltiopsis pygmaea* (J.Agardh) P.C.Silva & DeCew	N.S.	N	Y	N	[16]	
Gigartinales	Rhizophyllidaceae	*Portieria hornemannii* (Lyngbye) P.C.Silva	N.S.	N	Y	N	[16]	
Gigartinales	Solieriaceae	*Meristotheca* sp.1FP	N.L.	Y	Y	Y	This study	*rbc*L
Gigartinales	Solieriaceae	*Meristotheca procumbens* P.W.Gabrielson & Kraft	N.S.	N	Y	N	[16]	
Gigartinales	Solieriaceae	*Sarconema filiforme* (Sonder) Kylin	N.S.	N	Y	N	[16]	
Gigartinales	Solieriaceae	*Wurdemannia miniata* (Sprengel) Feldmann & Hamel	N.S.	N	Y	N	[16]	
Gracilariales	Gracilariaceae	*Gracilaria abbottiana* M.D.Hoyle	N.S.	N	Y	N	[16]	
Gracilariales	Gracilariaceae	*Gracilaria caudata* J.Agardh	N.R.	N	Y	Y	This study	*rbc*L
Gracilariales	Gracilariaceae	*Gracilaria flabelliformis* (P.Crouan & H.Crouan) Fredericq & Gurgel	N.R.	N	Y	Y	This study	*rbc*L
Gracilariales	Gracilariaceae	*Gracilaria isabellana* Gurgel, Fredericq & J.N.Norris	N.R.	N	Y	Y	This study	*rbc*L
Gracilariales	Gracilariaceae	*Gracilaria parvispora* I.A.Abbott	N.S.	N	Y	N	[16]	
Gracilariales	Gracilariaceae	*Gracilaria* sp.1FP	N.L.	Y	Y	Y	This study	*rbc*L
Halymeniales	Grateloupiaceae	*Grateloupia filicina* (J.V.Lamouroux) C.Agardh	CONF.	N	Y	Y	This study, [16,58]	*rbc*L, 28S
Halymeniales	Grateloupiaceae	*Grateloupia filiformis* Kützing	N.R.	N	Y	Y	This study	
Halymeniales	Grateloupiaceae	*Grateloupia hawaiiana* E.Y.Dawson	N.R.	N	Y	Y	This study	*rbc*L
Halymeniales	Grateloupiaceae	*Grateloupia phuquocensis* Tanaka & Pham-Hoàng Hô	N.S.	N	Y	N	[16]	
Halymeniales	Halymeniaceae	*Cryptonemia palmetta* (S.G.Gmelin) Woelkering, G.Furnari, Cormaci & McNeill	N.S.	N	Y	N	[16]	
Halymeniales	Halymeniaceae	*Halymenia actinophysa* M.Howe	N.S.	N	Y	N	[16]	
Halymeniales	Halymeniaceae	*Halymenia nukuhivensis* N’Yeurt & Payri	N.S.	N	Y	N	[16]	
Halymeniales	Tsengiaceae	*Tsengia abbottiana* (J.N.Norris & Bucher) J.N.Norris & Bucher	N.S.	N	Y	N	[16]	
Nemaliales	Galaxauraceae	*Actinotrichia fragilis* (Forsskål) Børgesen	CONF.	N	Y	Y	This study, [16]	*rbc*L
Nemaliales	Galaxauraceae	*Actinotrichia* sp.1FP	N.L.	Y	Y	Y	This study	*rbc*L
Nemaliales	Galaxauraceae	*Actinotrichia* sp.2FP	N.L.	Y	Y	Y	This study	*rbcL*
Nemaliales	Galaxauraceae	*Actinotrichia* sp.3FP	N.L.	Y	Y	Y	This study	*rbcL*
Nemaliales	Galaxauraceae	*Dichotomaria marginata* (J.Ellis & Solander) Lamarck	CONF.	N	Y	Y	This study, [16]	*rbc*L
Nemaliales	Galaxauraceae	*Dichotomaria obtusata* (J.Ellis & Solander) Lamarck	CONF.	N	Y	Y	This study, [16]	*rbc*L
Nemaliales	Galaxauraceae	*Galaxaura divaricata* (Linnaeus) Huisman & R.A.Townsend	CONF.	N	Y	Y	This study, [16]	*rbc*L
Nemaliales	Galaxauraceae	*Galaxaura filamentosa* R.C.Y.Chou	CONF.	N	Y	Y	This study, [16]	*rbc*L
Nemaliales	Galaxauraceae	*Galaxaura pacifica* Tanaka	N.L.	N	Y	Y	This study	*rbc*L
Nemaliales	Galaxauraceae	*Galaxaura rugosa* (J.Ellis & Solander) J.V.Lamouroux	CONF.	N	Y	Y	This study, [16]	*rbc*L
Nemaliales	Galaxauraceae	*Galaxaura* sp.1FP	N.L.	Y	Y	Y	This study	*rbc*L
Nemaliales	Galaxauraceae	*Tricleocarpa cylindrica* (J.Ellis & Solander) Huisman & Borowitzka	N.S.	N	Y	N	[16]	
Nemaliales	Liagoraceae	*Dermonema virens* (J.Agardh) Pedroche & Ávila Ortíz	N.S.	N	Y	N	[16]	
Nemaliales	Liagoraceae	*Ganonema papenfussii* (I.A.Abbott) J.M.Huisman, I.A.Abbott & A.R.Sherwood	N.S.	N	Y	N	[16]	
Nemaliales	Liagoraceae	*Ganonema* sp.1FP	N.S.	Y	Y	Y	Lin (unpubl.)	*rbc*L
Nemaliales	Liagoraceae	*Hommersandiophycus* sp.1FP	N.S.	Y	Y	Y	Lin (unpubl.)	*rbc*L
Nemaliales	Liagoraceae	*Liagora albicans* J.V.Lamouroux	N.S.	N	Y	N	[16]	
Nemaliales	Liagoraceae	*Liagora ceranoides* J.V.Lamouroux	N.S.	N	Y	N	[16]	
Nemaliales	Liagoraceae	*Liagora divaricata* C.K.Tseng	N.S.	N	Y	N	[16]	
Nemaliales	Liagoraceae	*Liagora* sp. Inedit	N.S.	Y	Y	N	[16]	
Nemaliales	Liagoraceae	*Liagora* sp.1FP	N.L.	Y	Y	Y	This study	*rbc*L
Nemaliales	Liagoraceae	*Titanophycus validus* (Harvey) Huisman, G.W.Saunders & A.R.Sherwood	N.S.	N	Y	N	[16]	
Nemaliales	Liagoraceae	*Trichogloea requienii* (Montagne) Kützing	CONF.	N	Y	Y	This study, [16]	*rbc*L
Nemaliales	Scinaiaceae	*Gloiophloea articulata* Weber Bosse	N.S.	N	Y	N	[16]	
Nemaliales	Yamadaellaceae	*Yamadaella caenomyce* (Decaisne) I.A.Abbott	N.S.	N	Y	N	[16]	
Nemastomatales	Nemastomataceae	*Predaea incraspeda* Kraft	N.S.	N	Y	N	[16]	
Nemastomatales	Nemastomataceae	*Predaea laciniosa* Kraft	N.S.	N	Y	N	[16]	
Nemastomatales	Nemastomataceae	*Predaea weldii* Kraft & I.A.Abbott	N.S.	N	Y	N	[16]	
Nemastomatales	Schizymeniaceae	*Platoma cyclocolpum* (Montagne) F.Schmitz	N.S.	N	Y	N	This study, [16]	
Nemastomatales	Schizymeniaceae	*Titanophora weberae* Børgesen	N.S.	N	Y	N	[16]	
Peyssonneliales	Peyssonneliaceae	*Agissea inamoena* (Pilger) Pestana, Lyra, Cassano & J.M.C. Nunes	N.S.	N	Y	N	[16]	
Peyssonneliales	Peyssonneliaceae	*Peyssonnelia bornetii* Boudouresque & Denizot	N.S.	N	Y	N	[16]	
Peyssonneliales	Peyssonneliaceae	*Peyssonnelia* sp.1FP	N.L.	Y	Y	Y	This study	*rbc*L
Peyssonneliales	Peyssonneliaceae	*Peyssonnelia* sp.2FP	N.L.	Y	Y	Y	This study	*rbc*L
Peyssonneliales	Peyssonneliaceae	*Peyssonnelia* sp.3FP	N.L.	Y	Y	Y	This study	*rbc*L
Peyssonneliales	Peyssonneliaceae	*Polystrata* sp.1FP	N.L.	Y	Y	Y	This study	*rbc*L
Plocamiales	Plocamiaceae	*Plocamium sandvicense* J.Agardh	N.S.	N	Y	N	[16]	
Rhodogorgonales	Rhodogorgonaceae	*Renouxia antillana* Fredericq & J.N.Norris	N.S.	N	Y	N	[16]	
Rhodymeniales	Champiaceae	*Champia compressa* Harvey	N.S.	N	Y	N	[16]	
Rhodymeniales	Champiaceae	*Champia parvula* (C.Agardh) Harvey	N.S.	N	Y	N	[16]	
Rhodymeniales	Champiaceae	*Champia* sp.1FP	N.L.	Y	Y	Y	This study	*rbc*L
Rhodymeniales	Champiaceae	*Champia* sp.2FP	N.L.	Y	Y	Y	This study	*rbc*L
Rhodymeniales	Champiaceae	*Champia* sp.3FP	N.L.	Y	Y	Y	This study	*rbc*L
Rhodymeniales	Champiaceae	*Champia vieillardii* Kützing	N.S.	N	Y	N	[16]	
Rhodymeniales	Champiaceae	*Coelothrix irregularis* (Harvey) Børgesen	CONF.	N	Y	Y	This study, [16]	*rbc*L
Rhodymeniales	Faucheaceae	*Gloiocladia* sp. inedit	N.S.	N	Y	N	[16]	
Rhodymeniales	Faucheaceae	*Gloioderma iyoense* Okamura	N.S.	N	Y	N	[16]	
Rhodymeniales	Hymenocladiaceae	*Asteromenia anastomosans* (Weber Bosse) G.W.Saunders, C.E.Lane, C.W.Schneider & Kraft	N.S.	N	Y	N	[16]	
Rhodymeniales	Hymenocladiaceae	*Asteromenia pseudocoalescens* (Sonder ex Kützing) J.Agardh	N.S.	N	Y	N	[16]	
Rhodymeniales	Hymenocladiaceae	*Asteromenia* sp.1FP	N.L.	Y	Y	Y	This study	*rbc*L
Rhodymeniales	Lomentariaceae	*Ceratodictyon intricatum* (C.Agardh) R.E.Norris	N.S.	N	Y	N	[16]	
Rhodymeniales	Lomentariaceae	*Ceratodictyon scoparium* (Montagne & Millardet) R.E.Norris	N.S.	N	Y	N	[16]	
Rhodymeniales	Lomentariaceae	*Ceratodictyon* sp.1FP	N.L.	Y	Y	Y	This study	*rbc*L
Rhodymeniales	Lomentariaceae	*Ceratodictyon variabile* (J.Agardh) R.E.Norris	N.S.	N	Y	N	[16]	
Rhodymeniales	Lomentariaceae	*Lomentaria corallicola* Børgesen	N.S.	N	Y	N	[16]	
Rhodymeniales	Rhodymeniaceae	*Botryocladia skottsbergii* (Børgesen) Levring	N.S.	N	Y	N	[16]	
Rhodymeniales	Rhodymeniaceae	*Botryocladia* sp.1FP	N.L.	Y	Y	Y	This study	*rbc*L
Rhodymeniales	Rhodymeniaceae	*Botryocladia* sp.2FP	N.L.	Y	Y	Y	This study	*rbc*L
Rhodymeniales	Rhodymeniaceae	*Chamaebotrys boergesenii* (Weber Bosse) Huisman	N.S.	N	Y	N	[16]	
Rhodymeniales	Rhodymeniaceae	*Chrysymenia kaernbachii* Grunow	N.S.	N	Y	?	[77]	
Rhodymeniales	Rhodymeniaceae	*Chrysymenia* sp.1FP	N.L.	Y	Y	Y	This study	*rbc*L
Rhodymeniales	Rhodymeniaceae	*Gloiosaccion brownii* Harvey	N.S.	N	Y	N	[16]	
Rhodymeniales	Rhodymeniaceae	*Halichrysis* cf. *H. concrescens* (J.Agardh) De Toni	N.S.	N	Y	N	[16]	
Rhodymeniales	Rhodymeniaceae	*Halichrysis* sp.1FP	N.L.	Y	Y	Y	This study	*rbc*L
Rhodymeniales	Rhodymeniaceae	*Halichrysis* sp.2FP	N.L.	Y	Y	Y	This study	*rbc*L
Rhodymeniales	Rhodymeniaceae	*Halopeltis australis* (J.Agardh) G.W.Saunders	N.S.	N	Y	N	[16]	
Rhodymeniales	Rhodymeniaceae	*Halopeltis cuneata* (Harvey) G.W.Saunders	N.S.	N	Y	N	[16]	
Rhodymeniales	Rhodymeniaceae	*Halopeltis* sp.1FP	N.L.	Y	Y	Y	This study	*rbc*L
Rhodymeniales	Rhodymeniaceae	*Rhodymenia corallina* (Bory) Greville	N.S.	N	Y	N	[16]	
Rhodymeniales	Rhodymeniaceae	*Rhodymenia leptophylla* J.Agardh	N.S.	N	Y	N	[16]	
Rhodymeniales	Rhodymeniaceae	*Rhodymenia* sp. Inedit	N.S.	N	Y	N	[16]	
Rhodymeniales	Rhodymeniaceae	*Rhodymenia* sp. Inedit	N.S.	N	Y	N	[16]	
Sporolithales	Sporolithaceae	*Sporolithon episoredion* (W.H.Adey, R.A.Townsend & Boykins) Verheij	N.S.	N	Y	N	[16]	
Sporolithales	Sporolithaceae	*Sporolithon ptychoides* Heydrich	N.S.	N	Y	N	[16]	
Stylonematales	Stylonemataceae	*Rhodaphanes* sp.1FP	N.L.	Y	Y	Y	This study	*rbc*L
Stylonematales	Stylonemataceae	*Stylonema alsidii* (Zanardini) K.M.Drew	N.S.	N	Y	N	[16]	
Bryopsidales	Boodleaceae	*Boodlea composita* (Harvey) F.Brand	N.S.	N	Y	N	[15]	
Bryopsidales	Boodleaceae	*Boodlea* sp.1FPa	N.L.	Y	Y	Y	This study	ITS
Bryopsidales	Boodleaceae	*Boodlea* sp.1FPb	N.L.	Y	Y	Y	This study	ITS
Bryopsidales	Boodleaceae	*Boodlea* sp.1FPc	N.L.	Y	Y	Y	This study	ITS
Bryopsidales	Bryopsidaceae	*Bryopsis pennata* J.V.Lamouroux	CONF.	N	Y	Y	This study	*rbc*L, *tuf*A
Bryopsidales	Bryopsidaceae	*Bryopsis pennata* var. *secunda* (Harvey) Collins & Hervey	CONF.	N	Y	Y	[15,77], This study	*rbc*L, *tuf*A
Bryopsidales	Bryopsidaceae	*Bryopsis plumosa* (Hudson) C.Agardh	N.S.	N	Y	N	[15]	
Bryopsidales	Bryopsidaceae	*Bryposis* sp.1FP	N.L.	Y	Y	Y	This study	*tuf*A
Bryopsidales	Bryopsidaceae	*Bryposis* sp.2FP	N.L.	Y	Y	Y	This study	*rbc*L, *tuf*A
Bryopsidales	Caulerpaceae	*Caulerpa bikinensis* W.R.Taylor	CONF.	N	Y	Y	This study, [15]	*tuf*A
Bryopsidales	Caulerpaceae	*Caulerpa chemnitzia* (Esper) J.V.Lamouroux.	CONF.	N	Y	Y	This study, [15]	*tuf*A
Bryopsidales	Caulerpaceae	*Caulerpa chemnitzia* var. *turbinata* (J.Agardh) Fernández-García & Riosmena-Rodríguez	N.S.	N	Y	N	[15]	
Bryopsidales	Caulerpaceae	*Caulerpa cupressoides* (Vahl) C.Agardh	CONF.	N	Y	Y	This study, [15]	*tuf*A
Bryopsidales	Caulerpaceae	*Caulerpa cupressoides* var. l*ycopodium* Weber Bosse	N.S.	N	Y	N	[15]	
Bryopsidales	Caulerpaceae	*Caulerpa cupressoides* var. *mamillosa* (Montagne) Weber Bosse	N.S.	N	Y	N	[15]	
Bryopsidales	Caulerpaceae	*Caulerpa nummularia* Harvey ex J.Agardh	CONF.	N	Y	Y	This study, [15]	*tuf*A
Bryopsidales	Caulerpaceae	*Caulerpa oligophylla* Montagne	N.L.	N	Y	Y	This study	*tuf*A
Bryopsidales	Caulerpaceae	*Caulerpa racemosa* (Forsskål) J.Agardh	CONF.	N	Y	Y	This study, [15]	*tuf*A
Bryopsidales	Caulerpaceae	*Caulerpa racemosa* var. *macrophysa* (Sonder ex Kützing) W.R.Taylor	N.L.	N	Y	Y	This study	*tuf*A
Bryopsidales	Caulerpaceae	*Caulerpa serrulata* (Forsskål) J.Agardh	CONF.	N	Y	Y	This study, [15]	*tuf*A
Bryopsidales	Caulerpaceae	*Caulerpa sertularioides* (S.G.Gmelin) M.Howe	CONF.	N	Y	Y	This study, [15]	*tuf*A
Bryopsidales	Caulerpaceae	*Caulerpa seuratii* Weber Bosse	N.S.	N	Y	N	[15]	
Bryopsidales	Caulerpaceae	*Caulerpa taxifolia* (M.Vahl) C.Agardh	CONF.	N	Y	Y	This study, [15]	*tuf*A
Bryopsidales	Caulerpaceae	*Caulerpa taxifolia* f. *tristichophylla* Svedelius	N.S.	N	Y	N	[15]	
Bryopsidales	Caulerpaceae	*Caulerpa urvilleana* Montagne	CONF.	N	Y	Y	This study, [15]	*tuf*A
Bryopsidales	Caulerpaceae	*Caulerpa verticillata* J.Agardh	CONF.	N	Y	Y	This study, [15]	*tuf*A
Bryopsidales	Caulerpaceae	*Caulerpa webbiana* Montagne	CONF.	N	Y	Y	This study, [15]	*tuf*A
Bryopsidales	Caulerpaceae	*Caulerpa webbiana* var. *pickeringii* (Harvey & Bailey) Eubank	N.S.	N	Y	N	[15]	*tuf*A
Bryopsidales	Codiaceae	*Codium arabicum* Kützing	CONF.	N	Y	Y	This study, [15]	*tuf*A
Bryopsidales	Codiaceae	*Codium geppiorum* O.C.Schmidt	CONF.	N	Y	Y	This study, [15]	*tuf*A
Bryopsidales	Codiaceae	*Codium lucasii* Setchell	N.L.	N	Y	Y	This study	*tuf*A
Bryopsidales	Codiaceae	*Codium mamillosum* Harvey	N.S.	N	Y	N	[15]	
Bryopsidales	Codiaceae	*Codium ovale* Zanardini	N.S.	N	Y	N	[77]	
Bryopsidales	Codiaceae	*Codium repens* P.Crouan & H.Crouan	N.L.	N	Y	Y	This study	*tuf*A
Bryopsidales	Codiaceae	*Codium saccatum* Okamura	N.S.	N	Y	N	[16]	
Bryopsidales	Codiaceae	*Codium sursum* Kraft & A.J.K.Millar	N.L.	N	Y	Y	This study	*tuf*A
Bryopsidales	Derbesiaceae	*Derbesia marina* (Lyngbye) Solier	N.S.	N	Y	N	[15]	
Bryopsidales	Dichotomosiphonaceae	*Avrainvillea amadelpha* (Montagne) A.Gepp & E.S.Gepp	N.L.	N	Y	Y	This study	*tuf*A
Bryopsidales	Dichotomosiphonaceae	*Avrainvillea calathina* Kraft & Olsen-Stojkovich	CONF.	N	Y	Y	[125]	*tuf*A
Bryopsidales	Dichotomosiphonaceae	*Avrainvillea erecta* (Berkeley) A.Gepp & E.S.Gepp	N.S.	N	Y	N	[15]	
Bryopsidales	Dichotomosiphonaceae	*Avrainvillea lacerata* J.Agardh	CONF.	N	Y	Y	This study, [15]	*tuf*A
Bryopsidales	Dichotomosiphonaceae	*Avrainvillea obscura* (C.Agardh) J.Agardh	N.S.	N	Y	N	[15]	
Bryopsidales	Dichotomosiphonaceae	*Avrainvillea ridleyi* A.Gepp & E.S.Gepp	N.S.	N	Y	N	[15]	
Bryopsidales	Dichotomosiphonaceae	*Avrainvillea* sp.1FP	N.L.	Y	Y	Y	[125]	*tuf*A
Bryopsidales	Dichotomosiphonaceae	*Avrainvillea spongiosa*	CONF.	N	Y	Y	[125]	*rbc*L, *tuf*A
Bryopsidales	Halimedaceae	*Chlorodesmis fastigiata* (C.Agardh) S.C.Ducker	N.S.	N	Y	N	[15]	
Bryopsidales	Halimedaceae	*Halimeda borneensis* W.R.Taylor	CONF.	N	Y	Y	This study, [15,64,65]	*tuf*A
Bryopsidales	Halimedaceae	*Halimeda discoidea* Decaisne	CONF.	N	Y	Y	This study, [15,64]	*tuf*A
Bryopsidales	Halimedaceae	*Halimeda distorta* (Yamada) Hillis-Colinvaux	CONF.	N	Y	Y	[15]	
Bryopsidales	Halimedaceae	*Halimeda distorta*.1 (Yamada) Hillis-Colinvaux	CONF.	N	Y	Y	This study, [64,66]	*tuf*A
Bryopsidales	Halimedaceae	*Halimeda distorta*.2 (Yamada) Hillis-Colinvaux	CONF.	N	Y	Y	[64,66]	*tuf*A
Bryopsidales	Halimedaceae	*Halimeda gracilis* Harvey ex J.Agardh	CONF.	N	Y	Y	[15,65,66,67]	*tuf*A
Bryopsidales	Halimedaceae	*Halimeda heteromorpha* N’Yeurt	CONF.	N	Y	Y	This study, [15,64,65]	*tuf*A
Bryopsidales	Halimedaceae	*Halimeda lacunalis* f. *lata* (W.R.Taylor) L.W.Hillis	N.S.	N	Y	N	[15]	
Bryopsidales	Halimedaceae	*Halimeda lacunalis* W.R.Taylor	N.L.	N	Y	Y	This study	*tuf*A
Bryopsidales	Halimedaceae	*Halimeda macroloba* Decaisne	CONF.	N	Y	Y	This study, [15]	*tuf*A
Bryopsidales	Halimedaceae	*Halimeda melanesica* Valet	CONF.	N	Y	Y	This study, [15]	*tuf*A
Bryopsidales	Halimedaceae	*Halimeda micronesica* Yamada	CONF.	N	Y	Y	This study, [15]	*tuf*A
Bryopsidales	Halimedaceae	*Halimeda minima* (W.R.Taylor) Hillis-Colinvaux	CONF.	N	Y	Y	This study, [15,66]	*tuf*A
Bryopsidales	Halimedaceae	*Halimeda opuntia* (Linnaeus) J.V.Lamouroux	CONF.	N	Y	Y	This study, [15,64,65,68]	*tuf*A
Bryopsidales	Halimedaceae	*Halimeda taenicola* W.R.Taylor	CONF.	N	Y	Y	This study, [15,64,65,67]	*tuf*A
Bryopsidales	Halimedaceae	*Rhipidosiphon javensis* Montagne	N.S.	N	Y	N	[15]	
Cladophorales	Anadyomenaceae	*Anadyomene saldanhae* A.B.Joly & E.C.Oliveira	N.L.	N	Y	Y	This study	
Cladophorales	Anadyomenaceae	*Microdictyon* sp.1FP	N.L.	Y	Y	Y	This study	28S
Cladophorales	Anadyomenaceae	*Microdictyon boergesenii* Setchell	N.L.	N	Y	Y	This study	28S
Cladophorales	Anadyomenaceae	*Microdictyon okamurae* Setchell	N.S.	N	Y	N	[15]	
Cladophorales	Anadyomenaceae	*Microdictyon* sp.	N.S.	N	Y	N	[15]	
Cladophorales	Anadyomenaceae	*Microdictyon umbilicatum* (Velley) Zanardini	CONF.	N	Y	Y	This study, [15]	28S
Cladophorales	Boodleaceae	*Cladophoropsis fasciculata* (Kjellman) Wille	N.S.	N	Y	N	[15]	
Cladophorales	Boodleaceae	*Phyllodictyon anastomosans* (Harvey) Kraft & M.J.Wynne	N.S.	N	Y	N	[15]	18S
Cladophorales	Boodleaceae	*Struveopsis* sp.	N.S.	N	Y	N	[15]	
Cladophorales	Cladophoraceae	*Chaetomorpha antennina* (Bory) Kützing	N.S.	N	Y	N	[15]	
Cladophorales	Cladophoraceae	*Chaetomorpha basiretrorsa* Setchell	N.S.	N	Y	N	[15]	
Cladophorales	Cladophoraceae	*Chaetomorpha fibrosa* (Kützing) Kützing	N.S.	N	Y	N	[15]	
Cladophorales	Cladophoraceae	*Chaetomorpha linum* (O.F.Müller) Kützing	N.S.	N	Y	N	[15]	
Cladophorales	Cladophoraceae	*Cladophora aokii* Yamada	N.S.	N	Y	N	[15]	
Cladophorales	Cladophoraceae	*Cladophora catenata* Kützing	CONF.	N	Y	Y	This study, [15]	28S
Cladophorales	Cladophoraceae	*Cladophora glomerata* (Linnaeus) Kützing	N.S.	N	Y	N	[77]	
Cladophorales	Cladophoraceae	*Cladophora goweri* A.H.S.Lucas	N.S.	N	Y	N	[77]	
Cladophorales	Cladophoraceae	*Cladophora patentiramea* (Montagne) Kützing	N.S.	N	Y	N	[15]	
Cladophorales	Cladophoraceae	*Cladophora sericea* (Hudson) Kützing	N.S.	N	Y	N	[15]	
Cladophorales	Cladophoraceae	*Cladophora sibogae* Reinbold	N.L.	N	Y	Y	This study	18S, 28S
Cladophorales	Cladophoraceae	*Cladophora socialis* Kützing	CONF.	N	Y	Y	This study, [15]	18S, 28S
Cladophorales	Cladophoraceae	*Cladophora vagabunda* (Linnaeus) Hoek	N.L.	N	Y	Y	This study	28S
Cladophorales	Cladophoraceae	*Lychaete feredayoides* (Kraft & A.J.K.Millar) M.J.Wynne	N.S.	N	Y	N	[15]	
Cladophorales	Cladophoraceae	*Lychaete herpestica* (Montagne) M.J.Wynne	N.S.	N	Y	N	[15]	
Cladophorales	Cladophoraceae	*Lychaete ohkuboana* (Holmes) M.J.Wynne	N.S.	N	Y	N	[15]	
Cladophorales	Cladophoraceae	*Lychaete* sp.1FP	N.L.	Y	Y	Y	This study	18S, 28S
Cladophorales	Cladophoraceae	*Pseudorhizoclonium africanum* (Kützing) Boedeker	N.S.	N	Y	N	[15]	
Cladophorales	Cladophoraceae	*Rhizoclonium riparium* (Roth) Harvey	N.S.	N	Y	N	[15]	
Cladophorales	Siphonocladaceae	*Boergesenia forbesii* (Harvey) Feldmann	N.S.	N	Y	N	[77]	
Cladophorales	Siphonocladaceae	*Dictyosphaeria cavernosa* (Forsskål) Børgesen	N.S.	N	Y	N	[15]	
Cladophorales	Siphonocladaceae	*Dictyosphaeria* sp.1FP	N.L.	Y	Y	Y	This study	28S
Cladophorales	Siphonocladaceae	*Dictyosphaeria* sp.2FP	N.L.	Y	Y	Y	This study	28S
Cladophorales	Siphonocladaceae	*Dictyosphaeria* sp.3FP	N.L.	Y	Y	Y	This study	28S
Cladophorales	Siphonocladaceae	*Dictyosphaeria* sp.4FP	N.L.	Y	Y	Y	This study	28S
Cladophorales	Siphonocladaceae	*Dictyosphaeria* sp.5FP	N.L.	Y	Y	Y	This study	28S
Cladophorales	Siphonocladaceae	*Dictyosphaeria versluysii* Weber Bosse	N.S.	N	Y	N	[15]	
Cladophorales	Siphonocladaceae	*Siphonocladaceae* sp.1FP	N.L.	Y	Y	Y	This study	18S, 28S
Cladophorales	Siphonocladaceae	*Siphonocladus tropicus* (P.Crouan & H.Crouan) J.Agardh	N.S.	N	Y	N	[15]	
Cladophorales	Valoniaceae	*Valonia aegagropila* C.Agardh	N.S.	N	Y	N	[15]	
Cladophorales	Valoniaceae	*Valonia fastigiata* Harvey ex J.Agardh	CONF.	N	Y	Y	This study, [15]	28S
Cladophorales	Valoniaceae	*Valonia macrophysa* Kützing	N.S.	N	Y	N	[15]	
Cladophorales	Valoniaceae	*Valonia nutrix* Kraft et A. Millar	N.S.	N	Y	N	[77]	
Cladophorales	Valoniaceae	*Valonia ventricosa* J.Agardh	CONF.	N	Y	Y	This study, [15]	18S, 28S
Cladophorales	Valoniaceae	*Valoniopsis pachynema* (G.Martens) Børgesen	N.S.	N	Y	N	[15]	
Dasycladales	Dasycladaceae	*Neomeris annulata* Dickie	N.S.	N	Y	N	[15]	
Dasycladales	Dasycladaceae	*Neomeris vanbosseae* M.Howe	N.S.	N	Y	N	[15]	
Dasycladales	Polyphysaceae	*Parvocaulis parvulus* (Solms-Laubach) S.Berger, Fettweiss, Gleissberg, Liddle, U.Richter, Sawitzky & Zuccarello	N.L.	N	Y	Y	This study, [15]	*tuf*A
Palmophyllales	Palmophyllaceae	*Verdigellas peltata* D.L.Ballantine & J.N.Norris	N.S.	N	Y	N	[15]	
Ulvales	Ulvaceae	*Ulva* cf. *prolifera* O.F.Müller	N.L.	N	Y	Y	This study	*tuf*A
Ulvales	Ulvaceae	*Ulva* cf. *tepida* Y.Masakiyo & S.Shimada	N.L.	N	Y	Y	This study	*tuf*A
Ulvales	Ulvaceae	*Ulva clathrata* (Roth) C.Agardh	N.S.	N	Y	N	[15]	
Ulvales	Ulvaceae	*Ulva compressa* Linnaeus	CONF.	N	Y	Y	[15]	
Ulvales	Ulvaceae	*Ulva flexuosa* Wulfen	N.S.	N	Y	N	[15]	
Ulvales	Ulvaceae	*Ulva intestinalis* Linnaeus	N.S.	N	Y	N	[77]	
Ulvales	Ulvaceae	*Ulva lactuca* Linnaeus	CONF.	N	Y	Y	This study, [15]	*tuf*A
Ulvales	Ulvaceae	*Ulva pilifera* (Kützing) Škaloud & Leliaert	N.L.	N	Y	Y	This study	*tuf*A
Ulvales	Ulvaceae	*Ulva rigida* C.Agardh	N.S.	N	Y	N	[15]	
Dictyotales	Dictyotaceae	*Dictyopteris delicatula* J.V.Lamouroux	CONF.	N	Y	Y	This study	*rbc*L
Dictyotales	Dictyotaceae	*Dictyopteris* sp.1FP	CONF.	N	Y	Y	This study, [14]	*rbc*L
Dictyotales	Dictyotaceae	*Dictyota acutiloba* J.Agardh	N.S.	N	Y	N	[14]	
Dictyotales	Dictyotaceae	*Dictyota bartayresiana* J.V.Lamouroux	CONF.	N	Y	Y	This study, [14]	*cox*1, *psb*A
Dictyotales	Dictyotaceae	*Dictyota ceylanica*1	CONF.	N	Y	Y	This study, [14,70]	*cox*1, *cox*3, *psb*A, 26S
Dictyotales	Dictyotaceae	*Dictyota friabilis* Setchell	CONF.	N	Y	Y	[14,71], This study	*cox*1, *cox*3, psaA
Dictyotales	Dictyotaceae	*Dictyota hamifera* Setchell	CONF.	N	Y	Y	[14,71], This study	*cox*1, *cox*3, nad1, *psb*A
Dictyotales	Dictyotaceae	*Dictyota rigida* De Clerck & Coppejans	CONF.	N	Y	Y	This study	*psb*A
Dictyotales	Dictyotaceae	*Dictyota* sp.10FP	N.L.	Y	Y	Y	This study	*cox*1, *psb*A
Dictyotales	Dictyotaceae	*Dictyota* sp.1FP	N.L.	Y	Y	Y	This study	*psb*A
Dictyotales	Dictyotaceae	*Dictyota* sp.2FP	N.L.	Y	Y	Y	This study	*cox*1, *psb*A
Dictyotales	Dictyotaceae	*Dictyota* sp.3FP	N.L.	Y	Y	Y	This study	*cox*1, *psb*A
Dictyotales	Dictyotaceae	*Dictyota* sp.4FP	N.L.	Y	Y	Y	This study	*cox*1, *psb*A
Dictyotales	Dictyotaceae	*Dictyota* sp.5FP	N.L.	Y	Y	Y	This study	*psb*A
Dictyotales	Dictyotaceae	*Dictyota* sp.6FP	N.L.	Y	Y	Y	This study	*psb*A
Dictyotales	Dictyotaceae	*Dictyota* sp.7FP	N.L.	Y	Y	Y	This study	*psb*A
Dictyotales	Dictyotaceae	*Dictyota* sp.8FP	N.L.	Y	Y	Y	This study	*cox*1, *psb*A
Dictyotales	Dictyotaceae	*Dictyota* sp.9FP	N.L.	Y	Y	Y	This study	*psb*A
Dictyotales	Dictyotaceae	*Lobophora abscondita* C.W.Vieira, Payri & De Clerck	CONF.	N	Y	Y	[13]	*cox*3, *rbc*L
Dictyotales	Dictyotaceae	*Lobophora aveiae* C.W.Vieira, Payri & M.Zubia	CONF.	N	Y	Y	[13,126]	*cox*3, *rbc*L
Dictyotales	Dictyotaceae	*Lobophora endeavouriae* C.W.Vieira	CONF.	N	Y	Y	[13]	*cox*3, *nad*1, *psb*A, *rbc*L
Dictyotales	Dictyotaceae	*Lobophora gambierensis* C.W.Vieira, Payri & M.Zubia	CONF.	Y	Y	Y	[13,126]	*cox*3, *psb*A, *rbc*L
Dictyotales	Dictyotaceae	*Lobophora huifetuae* C.W.Vieira & Payri	CONF.	N	Y	Y	[13,126]	*cox*3, *psb*A, *rbc*L
Dictyotales	Dictyotaceae	*Lobophora lamourouxii* Payri & C.W.Vieira	CONF.	N	Y	Y	[13]	*cox*3, *psb*A, *rbc*L
Dictyotales	Dictyotaceae	*Lobophora marquisensis* C.W.Vieira & Payri	CONF.	Y	Y	Y	[13,126]	*cox*3, *psb*A, *rbc*L
Dictyotales	Dictyotaceae	*Lobophora moanae* C.W.Vieira, Payri & M.Zubia	CONF.	N	Y	Y	[13,126]	*cox*3, *psb*A, *rbc*L
Dictyotales	Dictyotaceae	*Lobophora motuae* C.W.Vieira	CONF.	Y	Y	Y	[13]	*cox*3, *psb*A, *rbc*L
Dictyotales	Dictyotaceae	*Lobophora obscura* (Dickie) C.W.Vieira, De Clerck & Payri	CONF.	Y	Y	Y	[13]	*cox*3, *psb*A, *rbc*L
Dictyotales	Dictyotaceae	*Lobophora pacifica* (Setchell) C.W.Vieira, De Clerck & Payri	CONF.	N	Y	Y	[13,126]	*cox*3, *psb*A, *rbc*L
Dictyotales	Dictyotaceae	*Lobophora petila* C.W.Vieira, Payri & De Clerck	CONF.	N	Y	Y	[13]	*cox*3
Dictyotales	Dictyotaceae	*Lobophora polynesiensis* C.W.Vieira, Payri & M.Zubia	CONF.	Y	Y	Y	[13,126]	*cox*3, *psb*A, *rbc*L
Dictyotales	Dictyotaceae	*Lobophora providenceae* C.W.Vieira	CONF.	N	Y	Y	[13]	*cox*3, *psb*A, *rbc*L
Dictyotales	Dictyotaceae	*Lobophora puhoroae* C.W.Vieira & M.Zubia	CONF.	Y	Y	Y	[13]	*cox*3, *psb*A, *rbc*L
Dictyotales	Dictyotaceae	*Lobophora rechercheae* C.W.Vieira	CONF.	N	Y	Y	[13]	*cox*3
Dictyotales	Dictyotaceae	*Lobophora ruae* C.W.Vieira, A.D.R.N’Yeurt & M.Zubia	CONF.	N	Y	Y	[13,126]	*cox*3, *psb*A, *rbc*L
Dictyotales	Dictyotaceae	*Lobophora ruamataiae* C.W.Vieira & M.Zubia	CONF.	Y	Y	Y	[13]	*cox*3, *psb*A, *rbc*L
Dictyotales	Dictyotaceae	*Lobophora sawaikiae* C.W.Vieira	CONF.	Y	Y	Y	[13]	*cox*3, *psb*A, *rbc*L
Dictyotales	Dictyotaceae	*Lobophora setchellii* C.W.Vieira & Payri	CONF.	N	Y	Y	[13]	*cox*3, *psb*A, *rbc*L
Dictyotales	Dictyotaceae	*Lobophora societensis* C.W.Vieira, Payri & M.Zubia	CONF.	Y	Y	Y	[13,126]	*cox*3, *psb*A, *rbc*L
Dictyotales	Dictyotaceae	*Lobophora sonderi* C.W.Vieira, De Clerck & Payri	CONF.	N	Y	Y	[13,73]	*cox*3, *psb*A, *rbc*L
Dictyotales	Dictyotaceae	*Lobophora* sp.103	CONF.	Y	Y	Y	[13]	*psb*A, *rbc*L
Dictyotales	Dictyotaceae	*Lobophora* sp.123	CONF.	Y	Y	Y	[13]	*cox*3
Dictyotales	Dictyotaceae	*Lobophora* sp.128	CONF.	Y	Y	Y	[13]	*cox*3, *rbc*L
Dictyotales	Dictyotaceae	*Lobophora* sp.129	CONF.	Y	Y	Y	[13]	*cox*3, *rbc*L
Dictyotales	Dictyotaceae	*Lobophora* sp.132	CONF.	Y	Y	Y	[13]	*cox*3, *psb*A, *rbc*L
Dictyotales	Dictyotaceae	*Lobophora* sp.133	CONF.	Y	Y	Y	[13]	*cox*3, *psb*A, *rbc*L
Dictyotales	Dictyotaceae	*Lobophora* sp.134	CONF.	Y	Y	Y	[13]	*psb*A, *rbc*L
Dictyotales	Dictyotaceae	*Lobophora* sp.32	CONF.	Y	Y	Y	[13,126]	*cox*3, *psb*A, *rbc*L
Dictyotales	Dictyotaceae	*Lobophora* sp.70	CONF.	N	Y	Y	[13,126]	*rbc*L
Dictyotales	Dictyotaceae	*Lobophora taaroae* C.W.Vieira & M.Zubia	CONF.	Y	Y	Y	[13]	*cox*3, *psb*A, *rbc*L
Dictyotales	Dictyotaceae	*Lobophora tangaroae* C.W.Vieira, Payri & M.Zubia	CONF.	N	Y	Y	[13,73]	*cox*3, *psb*A, *rbc*L
Dictyotales	Dictyotaceae	*Lobophora tauruae* C.W.Vieira & M.Zubia	CONF.	Y	Y	Y	[13]	*cox*3, *psb*A, *rbc*L
Dictyotales	Dictyotaceae	*Lobophora tuamotuensis* C.W.Vieira	CONF.	Y	Y	Y	[13]	*cox*3, *psb*A, *rbc*L
Dictyotales	Dictyotaceae	*Lobophora tupaiae* C.W.Vieira, Payri & M.Zubia	CONF.	Y	Y	Y	[13,126]	*cox*3, *psb*A, *rbc*L
Dictyotales	Dictyotaceae	*Lobophora wakae* C.W.Vieira, Payri & M.Zubia	CONF.	N	Y	Y	[13,126]	*cox*3, *psb*A, *rbc*L
Dictyotales	Dictyotaceae	*Newhousia imbricata* Kraft, G.W.Saunders, I.A.Abbott & Haroun	CONF.	N	Y	Y	[21]	*cox*1, *cox*3, *psb*A, *rbc*L, SSU
Dictyotales	Dictyotaceae	*Padina boergesenii* Allender & Kraft	N.L.	N	Y	Y	This study	*cox*3
Dictyotales	Dictyotaceae	*Padina boryana* Thivy	CONF.	N	Y	Y	[14]	*cox*3
Dictyotales	Dictyotaceae	*Padina jonesii* Tsuda	N.L.	N	Y	Y	This study	*cox*3
Dictyotales	Dictyotaceae	*Padina melemele* I.A.Abbott & Magruder	N.S.	N	Y	N	[14]	
Dictyotales	Dictyotaceae	*Padina minor* Yamada	N.L.	N	Y	Y	This study	*cox*3
Dictyotales	Dictyotaceae	*Padina okinawaensis* Ni-Ni-Win, S.Arai & H.Kawai	N.L.	N	Y	Y	This study	*cox*3
Dictyotales	Dictyotaceae	*Padina pavonica* (Linnaeus) Thivy	N.S.	N	Y	N	[14]	
Dictyotales	Dictyotaceae	*Padina stipitata* Tanaka & Nozawa	N.S.	N	Y	N	[16]	
Dictyotales	Dictyotaceae	*Stypopodium* sp.1FP	CONF.	N	Y	Y	[14]	*rbc*L
Ectocarpales	Acinetosporaceae	*Feldmannia mitchelliae* (Harvey) H.-S.Kim	N.S.	N	Y	N	[14]	
Ectocarpales	Chordariaceae	*Cladosiphon novae-caledoniae* Kylin	N.S.	N	Y	N	[14]	
Ectocarpales	Scytosiphonaceae	*Chnoospora minima* (Hering) Papenfuss	CONF.	N	Y	Y	This study, [14]	*cox*3, *psb*A, *rbc*L
Ectocarpales	Scytosiphonaceae	*Colpomenia claytoniae* S.M.Boo, K.M.Lee, G.Y.Cho & W.Nelson	N.L.	N	Y	Y	This study	*cox*3, *rbc*L
Ectocarpales	Scytosiphonaceae	*Colpomenia sinuosa* (Mertens ex Roth) Derbès & Solier	CONF.	N	Y	Y	This study, [14]	*cox*3, *psb*A, *rbc*L
Ectocarpales	Scytosiphonaceae	*Hydroclathrus clathratus* (C.Agardh) M.Howe	N.S.	N	Y	N	[14]	
Ectocarpales	Scytosiphonaceae	*Hydroclathrus rapanuii* Santiañez, Macaya & Kogame	N.L.	N	Y	Y	This study	*cox*3, *rbc*L
Ectocarpales	Scytosiphonaceae	*Hydroclathrus tenuis* C.K.Tseng & Lu Baroen	N.L.	N	Y	Y	This study	*cox*3, *psb*A, *rbc*L
Ectocarpales	Scytosiphonaceae	*Hydroclathrus tilesii* (Endlicher) Santiañez & M.J.Wynne	N.L.	N	Y	Y	This study	*cox*3, *rbc*L
Ectocarpales	Scytosiphonaceae	*Hydroclathrus tumulis* Kraft & Abbott	N.S.	N	Y	N	[14]	
Ectocarpales	Scytosiphonaceae	*Manzaea minuta* (Santiañez & Kogame) Santiañez & Kogame	N.L.	N	Y	Y	This study	*cox*3, *psb*A, *rbc*L
Ectocarpales	Scytosiphonaceae	*Pseudochnoospora implexa* (J.Agardh) Santiañez, G.Y.Cho & Kogame	CONF.	N	Y	Y	This study, [14]	*cox*3, *psb*A, *rbc*L
Ectocarpales	Scytosiphonaceae	*Rosenvingea australis* Huisman, G.H.Boo & S.M.Boo	N.L.	N	Y	Y	This study	*cox*3, *rbc*L
Ectocarpales	Scytosiphonaceae	*Rosenvingea endiviifolia* (Martius) M.J.Wynne	CONF.	N	Y	Y	This study, [14]	*cox*3, *psb*A, *rbc*L
Ectocarpales	Scytosiphonaceae	*Rosenvingea* sp.1FP	N.L.	Y	Y	Y	This study	*cox*3, *psb*A, *rbc*L
Ectocarpales	Scytosiphonaceae	*Scytosiphon lomentaria* (Lyngbye) Link	N.S.	N	Y	N	[77]	
Fucales	Sargassaceae	*Sargassum echinocarpum* J.Agardh	CONF.	N	Y	Y	[14]	*cox*3, *rbc*L, ITS
Fucales	Sargassaceae	*Sargassum obtusifolium* J.Agardh	CONF.	N	Y	Y	[74]	*cox*3, *rbc*L, ITS
Fucales	Sargassaceae	*Sargassum pacificum* Bory	CONF.	N	Y	Y	[14]	*cox*3, *rbc*L, ITS
Fucales	Sargassaceae	*Spatoglossum* sp.1FP	N.L.	N	Y	Y	This study, [14]	*rbc*L
Fucales	Sargassaceae	*Turbinaria ornata* (Turner) J.Agardh	CONF.	N	Y	Y	[14]	*rbc*L, trnW-trnI, *tuf*A, ITS
Ralfsiales	Ralfsiaceae	*Neoralfsia expansa* (J.Agardh) P.-E.Lim & H.Kawai ex Cormaci & G.Furnari	N.S.	N	Y	N	[14]	
Scytothamnales	Asteronemataceae	*Asteronema breviarticulatum* (J.Agardh) Ouriques & Bouzon	N.S.	N	Y	N	[14]	
Sphacelariales	Sphacelariaceae	*Sphacelaria rigidula* Kützing	N.S.	N	Y	N	[14]	
Sphacelariales	Sphacelariaceae	*Sphacelaria tribuloides* Meneghini	N.S.	N	Y	N	[14]	
Tilopteridales	Cutleriaceae	*Cutleria irregularis* I.A.Abbott & Huisman	N.S.	N	Y	N	[14]	
Tilopteridales	Cutleriaceae	*Cutleria mollis* Allender & Kraft	N.S.	N	Y	N	[14]	
Chroococcales	Chroococcaceae	*Chroococcus membraninus* (Meneghini) Nägeli	N.S.	?	Y	N	[17]	
Chroococcales	Chroococcaceae	*Chroococcus minor* (Kützing) Nägeli	N.S.	?	Y	N	[17]	
Chroococcales	Chroococcaceae	*Chroococcus schizodermaticus* West	N.S.	?	Y	N	[17]	
Chroococcales	Chroococcaceae	*Chroococcus turgidus* (Kützing) Nägeli	N.S.	?	Y	N	[17]	
Chroococcales	Chroococcaceae	*Entophysalis conferta* (Kützing) Drouet & Daily	N.S.	?	Y	N	[17]	
Chroococcales	Chroococcaceae	*Entophysalis crustacea* (J.Agardh) Drouet et Daily	N.S.	?	Y	N	[17]	
Chroococcales	Chroococcaceae	*Entophysalis granulosa* Kützing	N.S.	?	Y	N	[17]	
Chroococcales	Chroococcaceae	*Entophysalis* sp.	N.S.	?	Y	N	[17]	
Chroococcales	Aphanothecaceae	*Aphanothece microscopica* Nägeli	N.S.	?	Y	N	[17]	
Chroococcales	Aphanothecaceae	*Aphanothece* sp.	N.S.	?	Y	N	[17]	
Chroococcales	Microcystaceae	*Aphanocapsa litoralis* Hansgirg	N.S.	?	Y	N	[17]	
Chroococcales	Microcystaceae	*Microcystis* sp.	N.S.	?	Y	N	[17]	
Chroococcidiopsidales	Aliterellaceae	*Chlorogloea* sp.	N.S.	?	Y	N	[17]	
Leptolyngbyales	Leptolyngbyaceae	*Heteroleibleinia erecta* (Gardner) Anagnostidis	N.S.	?	Y	N	[17]	
Leptolyngbyales	Leptolyngbyaceae	*Heteroleibleinia gardneri* (Geitler) Anagnostidis & Komárek	N.S.	?	Y	N	[17]	
Leptolyngbyales	Leptolyngbyaceae	*Heteroleibleinia infixa* (Frémy) Anagnostidis & Komárek	N.S.	?	Y	N	[17]	
Leptolyngbyales	Leptolyngbyaceae	*Heteroleibleinia willei* (Setchell & N.L.Gardner) Guiry & D.M.John	N.S.	?	Y	N	[17]	
Leptolyngbyales	Leptolyngbyaceae	*Pseudophormidium purpureum* (Gomont) Anagnostidis & Komárek.	N.S.	?	Y	N	[17]	
Leptolyngbyales	Trichocoleusaceae	*Schizothrix giuseppei* Drouet	N.S.	?	Y	N	[17]	
Leptolyngbyales	Trichocoleusaceae	*Schizothrix lacustris* A.Braun ex Gomont	N.S.	?	Y	N	[17]	
Leptolyngbyales	Trichocoleusaceae	*Schizothrix longiarticulata* Gardner	N.S.	?	Y	N	[17]	
Leptolyngbyales	Trichocoleusaceae	*Schizothrix minuta* (Forti) Geitler	CONF.	?	Y	Y	This study, [20]	16S
Leptolyngbyales	Trichocoleusaceae	*Schizothrix* sp.	N.S.	?	Y	N	[17]	
Leptolyngbyales	Trichocoleusaceae	*Schizothrix* sp.1FP	CONF.	?	Y	Y	This study	16S
Leptolyngbyales	Trichocoleusaceae	*Schizothrix telephoroides* Gomont	N.S.	?	Y	N	[17]	
Leptolyngbyales	Trichocoleusaceae	*Trichocoleus acutissimus* (N.L.Gardner) Anagnostidis	N.S.	?	Y	N	[17]	
Leptolyngbyales	Trichocoleusaceae	*Trichocoleus* sp.	CONF.	?	Y	Y	[20]	
Leptolyngbyales	Trichocoleusaceae	*Trichocoleus tenerrimus* (Gomont) Anagnostidis	N.S.	?	Y	N	[17]	
Nostocales	Aphanizomenonaceae	*Cylindrospermum licheniforme* Kützing ex Bornet & Flahault	N.S.	?	Y	N	[17]	
Nostocales	Aphanizomenonaceae	*Nodularia hawaiiensis* Tilden	N.S.	?	Y	N	[17]	
Nostocales	Aphanizomenonaceae	*Nodularia spumigena* Mertens ex Bornet & Flahault	N.S.	?	Y	N	[17]	
Nostocales	Aphanizomenonaceae	*Nodularia spumigena* var. *major* Bornet & Flahault	N.S.	?	Y	N	[17]	
Nostocales	Hapalosiphonaceae	*Fischerella ambigua* (Kützing ex Bornet & Flahault) Gomont	N.S.	?	Y	N	[17]	
Nostocales	Hapalosiphonaceae	*Hapalosiphon pumilis* Kirchner ex Bornet & Flahault	N.S.	?	Y	N	[17]	
Nostocales	Hapalosiphonaceae	*Mastigocoleus testarum* Lagerheim ex Bornet & Flahault	N.S.	?	Y	N	[17]	
Nostocales	Heteroscytonemataceae	*Heteroscytonema* sp.	CONF.	?	Y	Y	This study, [20]	16S
Nostocales	Nostocaceae	*Anabaena* sp.1	CONF.	?	Y	Y	This study, [20]	16S
Nostocales	Nostocaceae	*Anabaena* sp.2	CONF.	?	Y	Y	[20]	16S
Nostocales	Nostocaceae	*Anabaena torulosa* Lagerheim ex Bornet & Flahault	N.S.	?	Y	N	[17]	
Nostocales	Nostocaceae	*Desmonostoc muscorum* (Bornet & Flahault) Hrouzek & Ventura	N.S.	?	Y	N	[17]	
Nostocales	Nostocaceae	*Nostoc commune* Vaucher ex Bornet & Flahault	N.S.	?	Y	N	[17]	
Nostocales	Nostocaceae	*Nostoc ellipsosporum* Rabenhorst ex Bornet & Flahault	N.S.	?	Y	N	[17]	
Nostocales	Nostocaceae	*Nostoc linckia* Bornet ex Bornet & Flahault	N.S.	?	Y	N	[17]	
Nostocales	Nostocaceae	*Nostoc minutissimum* Kützing ex Bornet & Flahault	N.S.	?	Y	N	[17]	
Nostocales	Oscillatoriaceae	*Phormidium holdenii* (Forti) Branco, Sant’Anna, Azevedo & Sormus	N.S.	?	Y	N	[17]	
Nostocales	Rivulariaceae	*Calothrix aeruginea* Thuret ex Bornet & Flahault	N.S.	?	Y	N	[17]	
Nostocales	Rivulariaceae	*Calothrix clavata* West	N.S.	?	Y	N	[17]	
Nostocales	Rivulariaceae	*Calothrix confervicola* C.Agardh ex Bornet & Flahault	N.S.	?	Y	N	[17]	
Nostocales	Rivulariaceae	*Calothrix fusca* Bornet et Flahault	N.S.	?	Y	N	[17]	
Nostocales	Rivulariaceae	*Calothrix parietina* Thuret ex Bornet & Flahault	N.S.	?	Y	N	[17]	
Nostocales	Rivulariaceae	*Calothrix scopulorum* C.Agardh ex Bornet & Flahault	N.S.	?	Y	N	[17]	
Nostocales	Rivulariaceae	*Dichothrix hosfordii* Bornet	N.S.	?	Y	N	[17]	
Nostocales	Rivulariaceae	*Dichothrix rupicola* Collins	N.S.	?	Y	N	[17]	
Nostocales	Rivulariaceae	*Kyrtuthrix maculans* (Gomont) I.Umezaki	N.S.	?	Y	N	[17]	
Nostocales	Rivulariaceae	*Microchaete grisea* Thuret ex Bornet & Flahault	N.S.	?	Y	N	[17]	
Nostocales	Rivulariaceae	*Microchaete tapahiensis* Setchell	N.S.	?	Y	N	[17]	
Nostocales	Rivulariaceae	*Microchaete vitiensis* Askenasy	N.S.	?	Y	N	[17]	
Nostocales	Rivulariaceae	*Rivularia polyotis* Roth ex Bornet & Flahault	N.S.	?	Y	N	[17]	
Nostocales	Scytonemataceae	*Brachytrichia codii* Setchell	N.S.	?	Y	N	[17]	
Nostocales	Scytonemataceae	*Brachytrichia quoyi* Bornet & Flahault	N.S.	?	Y	N	[17]	
Nostocales	Scytonemataceae	*Scytonema coactile* Montagne ex Bornet & Flahault	N.S.	?	Y	N	[17]	
Nostocales	Scytonemataceae	*Scytonema guyanense* Bornet et Flahault	N.S.	?	Y	N	[17]	
Nostocales	Scytonemataceae	*Scytonema hoffmannii* C.Agardh ex Bornet & Flahault	N.S.	?	Y	N	[17]	
Nostocales	Scytonemataceae	*Scytonema ocellatum* Lyngbye ex Bornet & Flahault	N.S.	?	Y	N	[17]	
Nostocales	Scytonemataceae	*Scytonema polycystum* Bornet et Flahault	N.S.	?	Y	N	[17]	
Nostocales	Scytonemataceae	*Scytonema saleyeriense* Weber Bosse	N.S.	?	Y	N	[17]	
Nostocales	Scytonemataceae	*Scytonema varium* Kützing ex Bornet & Flahault	N.S.	?	Y	N	[17]	
Nostocales	Scytonemataceae	*Scytonema wolleanum* Forti	N.S.	?	Y	N	[17]	
Nostocales	Scytonemataceae	*Scytonematopsis crustacea (Thuret ex Bornet & Flahault) Kováčik & Komárek.*	N.S.	?	Y	N	[17]	
Nostocales	Scytonemataceae	*Scytonematopsis pilosa* (Bornet & Flahault) Umezaki & Watanabe	N.S.	?	Y	N	[17]	
Nostocales	Tolypothrichaceae	*Hassallia byssoidea Hassall ex Bornet & Flahault.*	N.S.	?	Y	N	[17]	
Oscillatoriales	Coleofasciculaceae	*Geitlerinema* sp.	CONF.	?	Y	Y	[20]	16S
Oscillatoriales	Coleofasciculaceae	*Geitlerinema* sp.1FP	CONF.	?	Y	Y	This study	16S
Oscillatoriales	Coleofasciculaceae	*Geitlerinema* sp.2FP	CONF.	?	Y	Y	This study	16S
Oscillatoriales	Microcoleaceae	*Arthrospira ardissonei* Forti	N.S.	?	Y	N	[17]	
Oscillatoriales	Microcoleaceae	*Arthrospira margaritae* (Frémy) Gomont ex Anagnostidis & Komárek	N.S.	?	Y	N	[17]	
Oscillatoriales	Microcoleaceae	*Arthrospira miniata* Gomont	CONF.	?	Y	Y	[20]	16S
Oscillatoriales	Microcoleaceae	*Blennothrix cantharidosma* (Gomont) Anagnostidis & Komárek	CONF.	?	Y	Y	This study, [17]	16S
Oscillatoriales	Microcoleaceae	*Blennothrix cantharidosma-B* (Gomont) Anagnostidis & Komárek	CONF.	?	Y	Y	[20]	16S
Oscillatoriales	Microcoleaceae	*Blennothrix cantharidosma-R* (Gomont) Anagnostidis & Komárek	CONF.	?	Y	Y	[20]	16S
Oscillatoriales	Microcoleaceae	*Blennothrix glutinosa* (Gomont) Anagnostidis & Komárek	CONF.	?	Y	Y	[17,20]	16S
Oscillatoriales	Microcoleaceae	*Blennothrix lyngbyacea* (Kützing ex Gomont) Anagnostidis & Komárek	N.S.	?	Y	N	[17]	
Oscillatoriales	Microcoleaceae	*Blennothrix major-B* (Holden) Anagnostidis & Komárek	CONF.	?	Y	Y	This study, [20]	16S
Oscillatoriales	Microcoleaceae	*Blennothrix major-GB* (Holden) Anagnostidis & Komárek	CONF.	?	Y	Y	[20]	
Oscillatoriales	Microcoleaceae	*Caldora penicillata* (Gomont) Engene, Tronholm & V.J.Paul	CONF.	?	Y	Y	This study, [20]	16S
Oscillatoriales	Microcoleaceae	*Caldora* sp.1FP	CONF.	?	Y	Y	This study	16S
Oscillatoriales	Microcoleaceae	*Coleofasciculus chthonoplastes* (Gomont) M.Siegesmund, J.R.Johansen & T.Friedl	N.S.	?	Y	N	[17]	
Oscillatoriales	Microcoleaceae	*Dapis pnigousa* Engene, Tronholm & V.J.Paul	CONF.	?	Y	Y	This study	16S
Oscillatoriales	Microcoleaceae	*Dapis* sp.1FP	CONF.	?	Y	Y	This study	16S
Oscillatoriales	Microcoleaceae	*Dapis* sp.2FP	CONF.	?	Y	Y	This study	16S
Oscillatoriales	Microcoleaceae	*Dapis* sp.3FP	CONF.	?	Y	Y	This study	16S
Oscillatoriales	Microcoleaceae	*Dapis* sp.4FP	CONF.	?	Y	Y	This study	16S
Oscillatoriales	Microcoleaceae	*Hydrocoleum coccineum* Gomont	CONF.	?	Y	Y	This study, [17,20]	16S
Oscillatoriales	Microcoleaceae	*Leibleinia epiphytica* (Hieronymus) Compère	N.S.	?	Y	N	[17]	
Oscillatoriales	Microcoleaceae	*Leibleinia gracilis* (Rabenhorst ex Gomont) Anagnostidis & Komárek	CONF.	?	Y	Y	This study, [17,20]	16S
Oscillatoriales	Microcoleaceae	*Limnospira maxima* (Setchell & N.L.Gardner) Nowicka-Krawczyk, Mühlsteinová & Hauer	N.S.	?	Y	N	[17]	
Oscillatoriales	Microcoleaceae	*Microcoleus paludosus* Gomont	N.S.	?	Y	N	[17]	
Oscillatoriales	Microcoleaceae	*Microcoleus* sp.	N.S.	?	Y	N	[17]	
Oscillatoriales	Microcoleaceae	*Neolyngbya* sp.1FP	CONF.	?	Y	Y	This study	16S
Oscillatoriales	Microcoleaceae	*Neolyngbya* sp.2FP	CONF.	?	Y	Y	This study	16S
Oscillatoriales	Microcoleaceae	*Okeania hirsuta* Engene, Paul, Byrum, Gerwick, Thor & Ellisman	CONF.	?	Y	Y	This study	16S
Oscillatoriales	Microcoleaceae	*Okeania plumata* N.Engene, V.J.Paul, T.Byrum, W.H.Gerwick, A.Thor & M.H.Ellisman	CONF.	?	Y	Y	This study	16S
Oscillatoriales	Microcoleaceae	*Okeania* sp.1FP	CONF.	?	Y	Y	This study	16S
Oscillatoriales	Microcoleaceae	*Okeania* sp.2FP	CONF.	?	Y	Y	This study	16S
Oscillatoriales	Microcoleaceae	*Plectonema mirabile* Thuret ex Gomont	N.S.	?	Y	N	[17]	
Oscillatoriales	Microcoleaceae	*Plectonema wollei* f. *gracilis* Frémy	N.S.	?	Y	N	[17]	
Oscillatoriales	Microcoleaceae	*Porphyrosiphon fuscus* Gomont ex Frémy	N.S.	?	Y	N	[17]	
Oscillatoriales	Microcoleaceae	*Symploca atlantica* Gomont	N.S.	?	Y	N	[17]	
Oscillatoriales	Microcoleaceae	*Symploca hydnoides* Kützing ex Gomont	CONF.	?	Y	Y	This study, [17,20]	16S
Oscillatoriales	Microcoleaceae	*Symploca kieneri* Drouet	N.S.	?	Y	N	[17]	
Oscillatoriales	Microcoleaceae	*Symploca laeteviridis* Gomont	N.S.	?	Y	N	[17]	
Oscillatoriales	Microcoleaceae	*Symploca muralis* Gomont	N.S.	?	Y	N	[17]	
Oscillatoriales	Microcoleaceae	*Symploca* sp.	CONF.	?	Y	Y	This study, [20]	16S
Oscillatoriales	Microcoleaceae	*Symploca* sp.1FP	CONF.	?	Y	Y	This study	16S
Oscillatoriales	Microcoleaceae	*Tychonema* sp.1FP	CONF.	?	Y	Y	This study	16S
Oscillatoriales	Oscillatoriaceae	*Kamptonema formosum* (Bory ex Gomont) Strunecký, Komárek & J.Smarda	N.S.	?	Y	N	[17]	
Oscillatoriales	Oscillatoriaceae	Kamptonema laetevirens (H.M.Crouan & P.L.Crouan ex Gomont) Strunecký, Komárek & J.Smarda	N.S.	?	Y	N	[17]	
Oscillatoriales	Oscillatoriaceae	*Lyngbya aestuari* Liebman ex Gomont	N.S.	?	Y	N	[17]	
Oscillatoriales	Oscillatoriaceae	*Lyngbya confervoides* C.Agardh ex Gomont	N.S.	?	Y	N	[17]	
Oscillatoriales	Oscillatoriaceae	*Lyngbya intermedia* Hansgirg	N.S.	?	Y	N	[17]	
Oscillatoriales	Oscillatoriaceae	*Lyngbya lutea* Gomont	N.S.	?	Y	N	[17]	
Oscillatoriales	Oscillatoriaceae	*Lyngbya majuscula* Harvey ex Gomont	CONF.	?	Y	Y	This study, [20]	16S
Oscillatoriales	Oscillatoriaceae	*Lyngbya martensiana* Meneghini ex Gomont	N.S.	?	Y	N	[17]	
Oscillatoriales	Oscillatoriaceae	*Lyngbya semiplena* J.Agardh ex Gomont	N.S.	?	Y	N	[17]	
Oscillatoriales	Oscillatoriaceae	*Lyngbya sordida* Gomont	N.S.	?	Y	N	[17]	
Oscillatoriales	Oscillatoriaceae	*Lyngbya* sp.1	CONF.	?	Y	Y	This study	16S
Oscillatoriales	Oscillatoriaceae	*Oscillatoria bonnemaisonii* P.Crouan & H.Crouan ex Gomont	CONF.	?	Y	Y	[17,20]	
Oscillatoriales	Oscillatoriaceae	*Oscillatoria corallinae* Gomont	N.S.	?	Y	N	[17]	
Oscillatoriales	Oscillatoriaceae	*Oscillatoria limosa* C.Agardh ex Gomont	N.S.	?	Y	N	[17]	
Oscillatoriales	Oscillatoriaceae	*Oscillatoria maraaensis* Setchell	N.S.	?	Y	N	[17]	
Oscillatoriales	Oscillatoriaceae	*Oscillatoria princeps* Vaucher ex Gomont	N.S.	?	Y	N	[17]	
Oscillatoriales	Oscillatoriaceae	*Oscillatoria sancta* f. *caldariorum* Elenkin	N.S.	?	Y	N	[17]	
Oscillatoriales	Oscillatoriaceae	*Oscillatoria sancta* f. *tenuis* (Woronichin) Elenkin	N.S.	?	Y	N	[17]	
Oscillatoriales	Oscillatoriaceae	*Oscillatoria sancta* Kützing ex Gomont	N.S.	?	Y	N	[17]	
Oscillatoriales	Oscillatoriaceae	*Oscillatoria* sp.1	CONF.	?	Y	Y	[20]	
Oscillatoriales	Oscillatoriaceae	*Oscillatoria* sp.2	CONF.	?	Y	Y	[20]	
Oscillatoriales	Oscillatoriaceae	*Oscillatoria* sp.3	CONF.	?	Y	Y	[20]	
Oscillatoriales	Oscillatoriaceae	*Oscillatoria tahitensis* Grunow ex Gomont	N.S.	?	Y	N	[17]	
Oscillatoriales	Oscillatoriaceae	*Phormidium ambiguum* Gomont	N.S.	?	Y	N	[17]	
Oscillatoriales	Oscillatoriaceae	*Phormidium breve* (Kützing ex Gomont) Anagnostidis & Komárek	N.S.	?	Y	N	[17]	
Oscillatoriales	Oscillatoriaceae	*Phormidium gardneri* Muzafarov	N.S.	?	Y	N	[17]	
Oscillatoriales	Oscillatoriaceae	*Phormidium laysanense* Lemmermann	N.S.	?	Y	N	[17]	
Oscillatoriales	Oscillatoriaceae	*Phormidium monile* Setchell et Gardner	N.S.	?	Y	N	[17]	
Oscillatoriales	Oscillatoriaceae	*Phormidium nigroviride* (Thwaites ex Gomont) Anagnostidis & Komárek	N.S.	?	Y	N	[17]	
Oscillatoriales	Oscillatoriaceae	*Phormidium papyraceum* Gomont	N.S.	?	Y	N	[17]	
Oscillatoriales	Oscillatoriaceae	*Phormidium* sp.1	CONF.	?	Y	Y	[20]	
Oscillatoriales	Oscillatoriaceae	*Phormidium* sp.2	CONF.	?	Y	Y	[20]	
Oscillatoriales	Oscillatoriaceae	*Phormidium* sp.3	CONF.	?	Y	Y	This study	16S
Oscillatoriales	Oscillatoriaceae	*Phormidium terebriforme* (C.Agardh ex Gomont) Anagnostidis & Komárek	N.S.	?	Y	N	[17]	
Pleurocapsales	Dermocarpellaceae	*Dermocarpa* sp.	N.S.	?	Y	N	[17]	
Pleurocapsales	Pleurocapsaceae	*Hyella caespitosa* Bornet et Flahault	N.S.	?	Y	N	[17]	
Pleurocapsales	Pleurocapsaceae	*Hyella* sp.	N.S.	?	Y	N	[17]	
Spirulinales	Spirulinaceae	*Spirulina rosea* P.Crouan & H.Crouan ex Gomont	CONF.	?	Y	Y	[20]	16S
Spirulinales	Spirulinaceae	*Spirulina* sp.1FP	CONF.	?	Y	Y	This study	16S
Spirulinales	Spirulinaceae	*Spirulina subsalsa* Oersted ex Gomont	CONF.	?	Y	Y	[17,20]	16S
Spirulinales	Spirulinaceae	*Spirulina tenerrima* Kützing ex Gomont	N.S.	?	Y	N	[17]	
Synechococcales	Leptolyngbyaceae	*Leptolyngbya calotrichoides* (Gomont) Anagnostidis & Komárek	N.S.	?	Y	N	[17]	
Synechococcales	Leptolyngbyaceae	*Leptolyngbya crossbyana* (Tilden) Anagnostidis & Komárek	CONF.	?	Y	Y	This study, [17]	16S
Synechococcales	Leptolyngbyaceae	*Leptolyngbya fragilis* (Gomont) Anagnostidis & Komárek	N.S.	?	Y	Y	[17]	
Synechococcales	Leptolyngbyaceae	*Leptolyngbya hendersonii* (Howe) Anagnostidis & Komárek	CONF.	?	Y	Y	This study, [20]	16S
Synechococcales	Leptolyngbyaceae	*Leptolyngbya mucicola* (Lemmermann) Anagnostidis & Komárek	N.S.	?	Y	N	[17]	
Synechococcales	Leptolyngbyaceae	*Leptolyngbya nostocorum* (Bornet ex Gomont) Anagnostidis & Komárek	N.S.	?	Y	N	[17]	
Synechococcales	Leptolyngbyaceae	*Leptolyngbya* sp.1	CONF.	?	Y	Y	This study, [20]	16S
Synechococcales	Leptolyngbyaceae	*Leptolyngbya* sp.2	CONF.	?	Y	Y	This study, [20]	16S
Synechococcales	Leptolyngbyaceae	*Leptolyngbya* sp.3	CONF.	?	Y	Y	This study	16S
Synechococcales	Leptolyngbyaceae	*Leptolyngbya* sp.4	CONF.	?	Y	Y	This study	16S
Synechococcales	Leptolyngbyaceae	*Leptolyngbya terebrans* (Bornet & Flahault ex Gomont) Anagnostidis & Komárek	N.S.	?	Y	N	[17]	
Synechococcales	Pseudanabaenaceae	*Pseudanabaena lonchoides* Anagnostidis	CONF.	?	Y	Y	[20]	16S
Synechococcales	Pseudanabaenaceae	*Pseudanabaena* sp.1FP	CONF.	?	Y	Y	This study	16S
Alismatales	Hydrocharitaceae	*Halophila decipiens* Ostenfeld	N.S.	N	Y	N	[127]	
Alismatales	Hydrocharitaceae	*Halophila ovalis* (R.Brown) Hooker f.	CONF.	N	Y	Y	This study, [AH]	ITS

CONF.: Confirmed with molecular data; N.L.: New lineage/species; N.S.: No sequence; N.R.: New record; Y: Yes; N: No; ?: unknown.

## Data Availability

The authors declare that all data supporting the findings of this study are available within the paper and its Appendix A. Phylogenetic trees generated for taxonomic identification are available upon request.

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
