# Peer review of "Marine Flora of French Polynesia: An Updated List Using DNA Barcoding and Traditional Approaches"

_biology, 2023, doi:10.3390/biology12081124_

Round 1

Reviewer 1 Report

This study contributes to the major revision of the French Polynesian marine flora by DNA barcording. By the species compositions authors nearly a two-fold increase of previous estimates.

The points below speak in favor of accepting the manuscript for the journal:

*The manuscript is clear, relevant for the field and well-structured.

*The abstract presented in the article characterizes the subject, reflects the purpose of the study, the main content and novelty of the article.

*The introduction contains historical and theoretical data according to modern literary sources.

 I have only several minor comments. Please, see them below:

Line 78 - …Tahitian algae was by [41] listing and illustrating 201 taxa… > …Tahitian algae was by Setchell [41] listing and illustrating 201 taxa…

Line 79 – Hollenberg [42-45] published on species of Rhodomelaceae

Lines 80 – 96 – The same situation, please, add the author of the corresponding study. Also line 302.

Line 150 – the second author > Kim M.-S.?

Lines 298 – 300 – Please italicize the latin taxa names.

Line 298 - B-Anabaena sp1 > B-Anabaena sp.1

Figures 1 and 2 – Please, check font style for the figures captions.

Line 329 – To my mind such a big table needs to be placed as the Suppl.file. Please also duplicate the head of the table on every page. The values in the columns on the right side of the table are unclear; please add notes below the table. Also, italicize cox, rbc, psb, tuf as usually required.

Line 416 - tufA is preferred for the Chlorophyta [81]. This statement is questionable. For Chlorophyta and Cyanobacteria it is better to use 18,16S rDNA genes (respectively) and/or ITS rDNA markers. The GenBank data, to my mind, are filled by these regions, but not tufA. This may be true for other groups where my experience is not as extensive. That’s why you could miss some precise identification during the study. So, speculation about the phylogenetic markers in 4.3.6. need to be revised or possibly greatly reduced.

Comment – I would like to see the phylogenetic trees. If authors think it will be a big paper. I’m agree.

Author Response

Dear Reviewer 1,

We would like to thank you for the time spent assessing our work. Thank you very much for your comments and suggestions. We address all the comments and corrections in the revised manuscript. We hope that this new version will meet your approval. 

Sincerely,

Mayalen Zubia and co-authors

Reviewer 2 Report

The presented MS is an important milestone in the study of the biodiversity of the marine flora of French Polynesia and thus the global biodiversity of oceanic plants.

The amount of barcoding work done by the Authors is impressive.

The tables in the appendix provide detailed PCR conditions for all markers used in the study, taxonomy and localization of identified samples. The resulting catalog of the marine flora of French Polynesia includes both specimens described by the authors and data from literary sources.

 Notes:

Table 1 does not explain the color scheme, and the abbreviations in the columns “Molecular confirmation” and “References”

Table S2 is titled as Table S1, S3 as S2.

Table S1 does not have a heading.

Author Response

Dear Reviewer 2,

We would like to thank you for the time spent assessing our work. Thank you very much for your comments and suggestions. We address all the comments and corrections in the revised manuscript. We hope that this new version will meet your approval. 

Sincerely,

Mayalen Zubia and co-authors

Reviewer 3 Report

Review for the paper "Marine flora of French Polynesia: an updated list using DNA barcoding and traditional approaches" by Christophe Vieira, Myung Sook Kim, Antoine De Ramon N'Yeurt, Claude Payri, Sophie D'Hondt, Olivier De Clerck, and Mayalen Zubia submitted to "Biology".

General comment.

The authors conducted a long-term study to determine the biodiversity of the marine flora in French Polynesia. They used both classical identification methods and modern DNA barcoding. As a result, the authors reassessed the species composition of the marine flora, including Alismatales, Cyanobacteria, Rhodophyta, Ochrophyta and Chlorophyta. The authors revised the checklist of the flora, which now consists of 702 species, including 119 Chlorophyta, 169 Cyanobacteria, 92 Ochrophyta, 320 Rhodophyta, and 2 seagrass species. The results of this study have significantly expanded the list of species. In addition, the authors provided a valuable DNA barcode reference library, which is very important for accurate species identification and future taxonomic and conservation studies. The authors used standard methods for sample collection and processing. The paper is well written and illustrated. I have only a few suggestions for improving the text.

Recommendations.

1) L 148. The authors are advised to include additional detail, primarily about the seasons that were investigated during the course of this study and the range of depth that was surveyed. These particulars, whilst currently discussed under the 'Discussion' section, would augment the 'Materials and Methods' section. Furthermore, it would be helpful to clarify the number of samples collected in each distinct zone.

2) In lines 298-300, the names of species ought to be italicized.

3) For Table 1, it is suggested to add a footnote explicating all abbreviations, inclusive of those utilized for references and color notations.

4) L 361, the authors should extend their discussion on spatial variation to incorporate a comparative study of the diversity patterns among the twelve islands that were sampled, or possibly among groups of these islands. There is a need here to investigate whether the habitat conditions display any spatial variation and if so, what is the contribution of this factor to the biodiversity of the indigenous flora.

5) The discussion may be further enriched by providing a detailed reflection on the authors' findings pertaining particularly to seasonal aspects.

Specific remarks.

L 17. Consider replacing “DNA  barcording” with “DNA  barcoding”

L 18. Consider deleting “from French Polynesia”

L 19. Consider replacing “119 species Chlorophyta” with “119 species of Chlorophyta”

L 112. Consider replacing “sequences were successful” with “sequences were successfully”

L 119. Consider replacing “The are several challenges” with “There are several challenges”

L 125. Consider replacing “high  quality  sequences” with “high-quality  sequences”

L 126. Consider replacing “while avoiding to put existing names on” with “while avoiding assigning existing names to”

L 137. Consider replacing “in Altantic European maerl beds” with “in Atlantic European maerl beds”

L 148. Consider replacing “between 2014 until 2023” with “between 2014 and 2023”

L 160. Consider replacing “based on most” with “based on the most”

L 165. Consider replacing “consisted in microscopic,” with “consisted of microscopic,”

L 177. Consider replacing “with date of collection. and deposited” with “with the date of collection and deposited”

L 188. Consider replacing “algal material were directly processed” with “algal material was directly processed”

L 198. Consider replacing “were as follow” with “were as follows”

L 213. Consider replacing “were carried using” with “were carried out using”

L 225. Consider replacing “allowed confirming that” with “allowed confirmation that”

L 229. Consider replacing “at higher taxonomic level” with “at a higher taxonomic level”

L 231. Consider replacing “label prior final publication” with “label prior to final publication”

L 236. Consider replacing “greater to 200 sequences)” with “greater than 200 sequences)”

L 247. Consider replacing “cluster of sequences” with “a cluster of sequences”

L 283. Consider replacing “diverged from closest” with “diverged from the closest”

L 291. Consider replacing “from previous checklist” with “from the previous checklist”

L 308. Consider replacing “169 Cyanobacteria” with “and 169 Cyanobacteria”

L 317. Consider replacing “At the level of a family” with “At the family level”

L 320. Consider replacing “endemism is of 11%” with “endemism is 11%”

L 458. Consider replacing “surveying and monitoring is needed” with “surveying and monitoring are needed”

L 493. Consider replacing “associated to its geographic distribution” with “associated with its geographic distribution”

MInor.

Author Response

Dear Reviewer 3,

We would like to thank you for the time spent assessing our work. Thank you very much for your comments and suggestions. We address all the comments and corrections in the revised manuscript. We hope that this new version will meet your approval. 

Sincerely,

Mayalen Zubia and co-authors

Reviewer 4 Report

This manuscript reports the 10-year effort to conduct a more comprehensive marine flora diversity study of French Polynesia. Not only it adds further insights into the marine floral diversity in French Polynesia, but also it exemplifies and well discusses the challenges and difficulties in using barcoding for marine flora diversity study. Despite some shortcomings (like the high percentage of no-matching with previous checklists), the study is overall well designed, executed, and written, and should have good general interest to those working on biodiversity and ecology.

Some suggestions for the authors to further improve:

1.       The authors used a 99% match as the cut-off value for ‘new species’, it is recommended to use other cut-off values (95%, 97%, and 98%) to test the sensitivity of the conclusion. It will be very value adding to this paper and future research by others.

2.       There is a discrepancy in the number of sequences generated in the different parts of this paper. A) Lanes258-259 states” Finally, a total of 1007 sequences …were generated from 2140 specimens B) Lanes403-404: “We were able to sequence 2140 specimens out of 1823 macroalgae, 316 cyanobacteria and 10 seagrasses,…”

Author Response

Dear Reviewer 4,

We would like to thank you for the time spent assessing our work. Thank you very much for your comments and suggestions. We address all the comments and corrections in the revised manuscript. We hope that this new version will meet your approval. 

Sincerely,

Mayalen Zubia and co-authors
